# Gaussian Mean Field Variational Inference can Overestimate Predictive Variance

**James Odgers** [1 2 3]  **Ben Riegler** [4 2 3]  **Siddharth Swaroop** [5]  **Vincent Fortuin** [1 2 3]

## Abstract

Mean Field Variational Inference (MFVI) is widely understood to underestimate posterior variance. By analysing conjugate Bayesian Linear Regression (BLR), we show that this characterization is incomplete: while MFVI underestimates the variance in parameter space, it can overestimate the predictive variance compared to the exact posterior. We show that if the MFVI posterior underestimates predictive variances in some directions, it necessarily overestimates them in others. Crucially, this overestimation occurs in directions where the training data concentrates. This leads to the surprising result that, for a test point drawn from the training distribution, MFVI's expected predictive variance exceeds that of the exact posterior. We demonstrate a pathological case of this effect, where the MFVI posterior fails to reduce predictive variance compared to the prior on ID data. We connect these results to the Cold Posterior Effect, arguing that varying the temperature can correct this overestimation, yielding predictions closer to those of the exact posterior. We validate our theory on synthetic and real-world regression tasks.

## 1. Introduction

Bayesian approaches to machine learning offer compelling advantages, including a principled approach to uncertainty estimation, and learning hyperparameters without overfitting (MacKay, 1992). Unfortunately, the full Bayesian posterior, $p(\theta|\mathcal{D})$, is usually intractable and needs to be approximated. One of the most popular approaches to approximate this posterior is Variational Inference (VI) (Jordan et al., 1999; Blei et al., 2017). Here, a family of distributions is proposed, $\mathcal{Q}$, from which the distribution $q^*(\theta)$ is optimized,

[1]University of Technology Nuremberg (UTN) [2]Helmholtz AI [3]Munich Center for Machine Learning (MCML) [4]Technische Universität München (TUM) [5]University College London (UCL). Correspondence to: James Odgers <james.odgers@utn.de>.

*Proceedings of the $43^{rd}$ International Conference on Machine Learning*, Seoul, South Korea. PMLR 306, 2026. Copyright 2026 by the author(s).

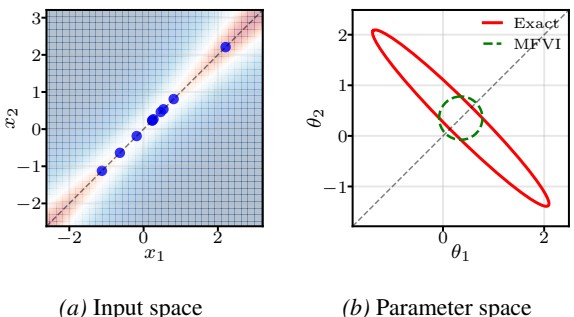

*(a)* Input space          *(b)* Parameter space

*Figure 1.* Although the MFVI posterior underestimates variance on average, it overestimates variance along the direction in which the data lie. Here we show this on a simple 2D linear regression, where the input data is restricted to lie on the subspace $x_1 = x_2$ (Figure 1a). We see that, for most of the input space, MFVI predictive uncertainty is less than the exact posterior predictive (blue region). At the same time, along the data direction, the opposite is true (red region). This is also the case when looking at the weight-space posterior (Figure 1b), with the exact posterior clearly taking up more volume, but also with the MFVI posterior having a larger variance along the crucial $\theta_1 = \theta_2$ subspace.

which minimises the reverse KL divergence to the true posterior, $q^*(\theta) = \arg\min_{q \in Q} D_{KL}(q(\theta)||p(\theta|\mathcal{D}))$. Practitioners often use Mean Field Variational Inference (MFVI), in which the variational posterior factorises across parameters, $q(\theta) = \prod_{p=1}^{P} q_p(\theta_p)$. In practice, the marginal distributions are often restricted to be Gaussian, so $q_p(\theta_p) = N(\mu_p, \sigma_p^2)$. We study this popular setting in this paper.

MFVI is typically considered to *underestimate* the uncertainty of the posterior (Minka, 2005; Turner and Sahani, 2011; Blei et al., 2017). This is because it is a mode-fitting approximation in parameter space: the approximate posterior avoids assigning probability mass to regions with low true probability, thereby underestimating the variance in parameter space (Margossian and Saul, 2023). However, in this paper, we argue that this picture is incomplete once the goal is prediction rather than parameter inference. In particular, the predictions from an MFVI posterior can *overestimate* uncertainty on In-Distribution (ID) data.

**Illustrative example (Figure 1).**   Figure 1a shows some simple training data restricted to the subspace $x_1 = x_2$, and Figure 1b shows the exact posterior and MFVI posterior (in parameter space) for this linear regression example. The

highly uneven, non-axis-aligned covariance of the training inputs produces a posterior with a strong covariance between $\theta_1$ and $\theta_2$ in the parameter space. We can see how MFVI appears to underestimate the variance in general: its mass is over a smaller region of the parameter space than the mass of the exact posterior. However, in this example, we are not interested in the parameters learned by the model; instead, we care about the predictions made on test data. A realistic assumption is often that future data points are similar to training data points, such that $x_1 = x_2$. If this is the case, the only direction in parameter space relevant for predictions is $\theta_1 = \theta_2$, as illustrated by the dashed line in Figure 1b. In this direction, the MFVI posterior *overestimates* the variance compared to the true posterior. Our main result, given in Theorem 3.7, is to show that the overestimation of predictive variance in MFVI in Bayesian Linear Regression (BLR) is very general.

**A Cold Posterior Effect without model mismatch.** The Cold Posterior Effect (CPE) from Bayesian Deep Learning (BDL) (Wenzel et al., 2020), describes an effect where artificially reducing the uncertainty of an approximate posterior improves predictive performance. In this work, we add another justification for why we may expect the CPE: lowering the temperature corrects for overestimated predictive variance from MFVI on in-distribution data, producing predictions which match the exact posterior predictions more closely. We demonstrate that the reverse effect is true for Out-Of-Distribution (OOD) data, where a warm posterior matches the exact posterior predictive distribution more closely and performs better on predictive tasks.

**Contributions.** In this paper, our main contributions are:

- We provide a theoretical analysis of conjugate Bayesian Linear regression, showing that the MFVI posterior overestimates the exact posterior variance along the first principal component of the training data, and that the average predicted variance at the training data points is higher for MFVI than the exact posterior.

- We theoretically and empirically examine the predictive performance of MFVI on a pathological setup and demonstrate that in the high-dimensional limit, the MFVI posterior can fail to reduce the predictive variance at all compared to the prior.

- We show theoretically and empirically that both a Cold Posterior Effect and Warm Posterior Effect can occur in linear regression models approximated with MFVI, and argue that both of these effects can be understood as scaling the predictive variance of the MFVI approximation to better match the exact posterior predictions.

## 2. Related Work

**Variational Inference (VI).** VI research has mostly focused on the approximation quality in parameter space, not the approximation quality on predictions (see Blei et al. (2017) for an example of this). One line of related research is Functional VI (Sun et al., 2019; Burt et al., 2021; Cinquin and Bamler, 2025), to allow more interpretable priors, such as GPs, to be applied to BNNs. In our linear regression setting, there is a one-to-one mapping between functions and parameters, so there is no difference in the functional and parametric variational objectives (see Proposition 3, Burt et al., 2021). A functional approach to VI is also common in Gaussian Processes (Titsias, 2009) to improve computational efficiency. We do not consider these approaches in this paper, as these approaches are equivalent to different variational families in parameter space which we do not study here. The paper closest to the theoretical analysis we provide is Margossian and Saul (2023), who studied the theoretical properties of the Gaussian MFVI posterior when the exact posterior is a Multivariate Gaussian. Crucially, this current work focuses on the properties of the predictions of VI, whereas Margossian and Saul (2023) focus on the properties in parameter space.

**Cold Posterior Effect (CPE).** There are several works proposing explanations for the CPE, showing how this effect can naturally arise from data augmentation (Izmailov et al., 2021; Nabarro et al., 2022; Bachmann et al., 2022), misspecified likelihoods (Adlam et al., 2020; Aitchison, 2021), misspecified priors (Fortuin et al., 2022; Kapoor et al., 2022; Marek et al., 2024), or a combination of effects (Zeno et al., 2020; Noci et al., 2021). We do not disagree with these causes; our paper provides an additional novel explanation, in which the model has a well-specified likelihood and prior, and no data augmentation is used. Instead, we propose that the CPE allows the MFVI approximation to more closely match the exact posterior predictions for certain tasks. Zhang et al. (2024) argue that the CPE is correcting for underconfident predictions, which we agree with and extend by providing a mechanistic understanding of this effect with MFVI. Many MFVI algorithms in deep learning temper their likelihood function, but do not provide any theoretical analysis or reasoning (e.g., Osawa et al., 2019; Ashman et al., 2022). We became aware of contemporaneous work by Harvey et al. (2025) showing that the standard MFVI-ELBO objective produces underconfident predictions, though their focus is on hyper-parameter learning, which we hold fixed.

**GenBayes.** Generalized Bayes centres around the idea that traditional Bayesian inference methods may not give models the best predictive properties; a tempering strategy (Aitchison, 2021), in which the likelihood and the prior regularisation terms of the VI objective are weighted differ-

ently, can produce better predictions (Pitas and Arbel, 2022; McLatchie et al., 2025). Our work is distinct from tempered objectives. Instead, we consider an objective approximating an exact cold posterior, which cannot be written with a tempered objective, as discussed in detail in Section 4.

## 3. MFVI Overestimates Predictive Uncertainty for the Empirical Distribution

**Problem Statement.** Assume we have a model where inputs $x \in \mathbb{R}^P$ and outputs $y \in \mathbb{R}$ are related by the linear predictor $\theta \in \mathbb{R}^P$, such that

$$y = \theta^\top x + \epsilon, \quad \epsilon \in \mathbb{R} \sim N(0, \sigma^2). \quad (1)$$

We assume that we have a training dataset of the form $\mathcal{D} = \{x_n, y_n\}_{n=1}^N = \{X \in \mathbb{R}^{N \times P}, Y \in \mathbb{R}^N\}$ drawn from this distribution. We will call the empirically observed distribution of inputs $\hat{p}(x) = \frac{1}{N} \sum_{n=1}^N \delta(x - x_n)$, and assume that the training data was mean-centred, so $\mathbb{E}_{\hat{p}(x)}[x] = 0$. Throughout this paper, we will assume that the prior, $\theta \sim N(0, \alpha^{-1}\mathbb{I})$, and likelihood, $\sigma^2$, are well specified and known to us.

**True Posterior.** In this work, we are interested in comparing to the true posterior

$$p(\theta|\mathcal{D}) = N(\mu, \Sigma), \quad (2)$$

$$\Sigma = \left(\frac{1}{\sigma^2} X^\top X + \alpha\mathbb{I}\right)^{-1}, \ \mu = \Sigma\left(\frac{1}{\sigma^2} X^\top Y\right).$$

Specifically, we are interested in comparing the predictions for the output, $y$, for a given test point, $x$, from the true posterior and approximate posteriors.

**Diagonalised True Posterior Covariance.** We will work with the diagonalised covariance matrix. For this, we define $\Sigma = V\Delta V^\top$, where $\Delta = \text{diag}(\delta_1, \ldots, \delta_P), \delta_p \in \mathbb{R}^+$, is a diagonal matrix and the columns of $V = [v_1, \ldots, v_P], v_p \in \mathbb{R}^P$, form a complete, orthonormal basis. We will use the expression

$$\Sigma = \sum_{q=1}^P \delta_q v_q v_q^\top. \quad (3)$$

**The variational posterior.** We are interested in finding an approximate posterior $q(\theta) = N(m, S)$ by minimising the reverse Kullback-Leibler (KL) divergence to the true posterior. For our linear regression model, the true posterior is Gaussian, so the objective to be minimised is

$$D_{KL}(q(\theta)||p(\theta|\mathcal{D})) = \frac{1}{2}\big[(\mu - m)^\top \Sigma^{-1}(\mu - m)$$
$$+ Tr(\Sigma^{-1}S) - \log\det S + \log\det\Sigma - P\big]. \quad (4)$$

This objective is minimised by the true posterior; however, this is commonly not available for computational reasons.

Instead, the variational posterior is constrained to factorise over its $P$ dimensions. In the Multivariate Gaussian case, this requires constraining the variational posterior to be diagonal, which is equivalent to fixing the eigenbasis to be axis-aligned. This allows us to write

$$S = \sum_{p=1}^P d_p e_p e_p^\top, \quad (5)$$

where $e_p$ are the canonical basis vectors which contain a single 1 at position $p$ and are zero otherwise.

In this form, the goal of fitting the variational posterior is given by finding the optimal parameters of $m^*$ and $\{d_p^*\}_{p=1}^P$. The optimal parameters for these, formalised in the following two lemmas, can be found as functions of the posterior mean, covariance eigenvalues, and the inner product of the orthonormal bases of the posteriors, $v_q^T e_p$. The proofs for these can be found in Appendix A.1.

**Lemma 3.1** (Optimal variational mean). *For any positive definite posterior covariance, the optimal mean is the mean of the true posterior:*

$$m^* = \mu. \quad (6)$$

**Lemma 3.2** (Optimal variational posterior eigenvalues are harmonic means). *The optimal variational covariance, $S^*$, is given by the condition*

$$\frac{1}{d_p^*} = e_p^\top \Sigma^{-1} e_p = \sum_{q=1}^P \frac{w_{pq}}{\delta_q} \quad \forall \, p \in \{1, \ldots P\}, \quad (7)$$

*where $w_{pq} = (v_q^T e_p)^2$ and $\sum_{q=1}^P w_{pq} = 1 \, \forall \, p \in \{1, \ldots, P\}$.*

While the values of the optimal variational posterior are well known in the literature (Turner and Sahani, 2011; Margossian and Saul, 2023), we are unaware of prior work describing the variational posterior eigenvalues as a weighted harmonic mean of the eigenvalues of the exact posterior. Later, this will be important for our theoretical arguments.

For a test input location, $x$, we are interested in comparing the predictive distributions excluding aleatoric uncertainty, which are given by

$$p(f|\mathcal{D}) = N(\mu^\top x, x^\top \Sigma x),$$
$$q(f) = N(m^{*\top} x, x^\top S^* x) \quad (8)$$

for the exact and MFVI posteriors, respectively. As the optimal variational and exact posterior means are identical, the only difference in these distributions is in the variance of the predictions, which is where we focus from now on.

### 3.1. Theoretical Analysis of Predictions from MFVI

This section proves theoretical results about the relationship between predictions from the MFVI and the exact posterior. We start by confirming the traditional story that MFVI underestimates variance, *i.e.* is overconfident, showing this is the case when the test input is isotropic. After this, we focus on the ways in which this traditional story is misleading. First, we show that there must be at least one direction in which the MFVI overestimates variance and, for spherical priors, this is the direction of maximum variance of the training data. After this, we show that the degree of over- and underestimation of the predictive variance is coupled together. Finally, we give our main result: MFVI overestimates variance, *i.e.* is underconfident, for test points similar to the training data. Formal proofs of all our theorems are in Appendix A.

**MFVI predictions underestimate uncertainty for isotropic test points.** We begin by considering the setting where the test distribution is isotropic, $x \sim N(0, \mathbb{I}_P)$. This corresponds to the assumption that test points are equally likely to arrive from any direction in the input space, a setting in which we would expect MFVI's tendency to underestimate variance to manifest clearly. Indeed, in this setting, we recover the standard result that MFVI is overconfident.

To see this, we note that the expected predictive variance under an isotropic test distribution is simply the trace of the posterior covariance:

$$\mathbb{E}_{x \sim N(0, \mathbb{I}_P)} \left[ x^\top A x \right] = \text{Tr}(A) \tag{9}$$

for any matrix $A$. The following lemma then follows directly from proving that the trace of the exact posterior is greater than the MFVI posterior, which we do in Lemma A.3.

**Lemma 3.3.** *For a test point $x \sim N(0, \mathbb{I}_P)$, the predictive variances of the MFVI posterior and exact posterior are governed by*

$$\mathbb{E}_{x \sim N(0, \mathbb{I}_P)} \left[ x^\top \Sigma x \right] \geq \mathbb{E}_{x \sim N(0, \mathbb{I}_P)} \left[ x^\top S^* x \right]. \tag{10}$$

This result aligns with the standard understanding that MFVI underestimates uncertainty. However, the assumption of isotropic test data is often unrealistic: in practice, we expect test points to follow a similar distribution to the training data. We now turn to this more realistic setting.

**MFVI overestimates uncertainty along the first principal component of the training data.** The first result we use to argue that MFVI overestimates uncertainty is to show that there is a direction in the input space where, if MFVI and exact posterior predictions are not equal, the MFVI posterior overestimates the predictive uncertainty.

This direction is the span of the eigenvector of the minimum eigenvalue of the true posterior, that is, for a test point $\tilde{x} \in \text{span}(v_{q^*})$ where $\delta_{min} = v_{q^*}^\top \Sigma v_{q^*}$.

**Lemma 3.4** (Overestimation of predictive variance in one direction of the input space). *For points in the input space defined by $\tilde{x} \in span(v_{q^*})$, the predictive variance for the exact posterior and the MFVI posterior are governed by the inequality*

$$\tilde{x}^\top S^* \tilde{x} \geq \tilde{x}^\top \Sigma \tilde{x}. \tag{11}$$

The intuition for this is that the exact posterior predictive variance will be $\delta_{min}$, which, because the eigenvalues of the MFVI posterior are a harmonic mean of the exact eigenvalues, must be lower than the lowest eigenvalue and the lowest predictive variance of the MFVI posterior.

In the common case of spherical priors, where the prior variance is given by $\alpha^{-1}\mathbb{I}$, the relative size of the posterior eigenvalues are entirely determined by the variance of the training data. This means that the direction in which this underestimation occurs is of great significance: it is the first principal component of the training data. More formally we can state the following theorem.

**Theorem 3.5.** *Consider the conjugate Bayesian linear model of Equation* (1) *with a spherical prior $\theta \sim N(0, \alpha^{-1}\mathbb{I})$, and let $w$ denote the first principal component of the training inputs $X$. Then for any test point $\tilde{x} \in span(w)$, the predictive variances of the MFVI and exact posteriors satisfy*

$$\tilde{x}^\top S^* \tilde{x} \geq \tilde{x}^\top \Sigma \tilde{x}. \tag{12}$$

The proof follows from noting that, for spherical priors, the exact posterior shares an eigenbasis with the gram matrix of the training data, $X^\top X$, so $w = v_{q^*}$ and Lemma 3.4 can be applied to give the desired result.

**MFVI under- and overestimates uncertainties similarly.** We can take the above result and strengthen it by showing that the degree of overestimation and underestimation of the predictive density are linked. To do this, we introduce the ratio of the predictive variances at test point $x$,

$$R(x) = \frac{x^\top S^* x}{x^\top \Sigma x}. \tag{13}$$

$R(x) > 1$ implies an overestimated uncertainty and $R(x) \leq 1$ implies underestimated uncertainty. We can now consider the average value of this new quantity across the complete eigenbasis of the exact posterior, and we will find the following surprising result.

**Lemma 3.6** (Calibrated MFVI). *Consider the set of eigenvectors of the true posterior $\{v_q\}_{q=1}^P$. For these points*

$$\frac{1}{P} \sum_{q=1}^P R(v_q) = 1. \tag{14}$$

While we are unable to provide an intuitive explanation for why this is the case, the result can be found by first showing that $\text{Tr}\left(\Sigma^{-1}S^*\right) = P$, as we do in Lemma A.4, and then performing straightforward algebra.

This constraint, that the average value of $R(x)$ is constant across eigenvectors of the true posterior, can cause problems for predicting in-distribution data. In particular, in high-dimensional settings, where many eigenvector directions have underestimated variance, the degree of overestimation in other directions must be large to compensate. We demonstrate an extreme version of this in Section 5.1.

**MFVI overestimates predictive variance for data with the empirical covariance.** So far, we have shown the following results: from Lemma 3.3 we can expect that there are a lot of directions in the input where the variance is underestimated, from Lemma 3.4 we can expect that for spherical priors the highest variance of the training data is underconfident, and from Lemma 3.6 we know that the degrees of this overestimation and underestimation are coupled in some way. These results provide some intuition for our main theorem, which says that, for BLR problems with a spherical prior, MFVI overestimates predictive uncertainty for test data points distributed according to the empirical distribution of the training data, $\hat{p}(x)$.

**Theorem 3.7.** *For test data points distributed according to the empirical distribution $x \sim \hat{p}(x)$, the difference in the expected predicted variance of the MFVI and exact posterior is given by*

$$\mathbb{E}_{x\sim\hat{p}(x)}\left[x^\top\Sigma x - x^\top S^* x\right] \leq 0. \qquad (15)$$

In our opinion, this is a pretty remarkable result: while MFVI can in most senses be considered to underestimate the epistemic uncertainty, in the very important case of predicting on test data which shares covariance with the training data, MFVI overestimates uncertainty.

## 4. Cold MFVI posterior predictions can match Bayesian predictions

**The Cold Posterior Objective for MFVI.** Cold posteriors describe exponentiated posteriors, $p_T(\theta|\mathcal{D}) \propto p(\mathcal{D}|\theta)^{1/T}p(\theta)^{1/T}$, where $T < 1$. This exponentiation corresponds to sharpening the exact posterior, so more of the mass is concentrated around the Maximum A Posteriori (MAP) estimate. If, rather than minimising the KL divergence to the exact posterior, we minimise the KL divergence to the cold posterior, we get a variational posterior

$$q_T^*(\theta) = \min_{q(\theta)\in Q} D_{KL}\left(q(\theta)||p_T(\theta|\mathcal{D})\right). \qquad (16)$$

We refer to this solution as the $T$-MFVI posterior. For the specific conjugate case where the prior has the form $N(0, \alpha^{-1}\mathbb{I}_P)$ and the likelihood has the form $N(y|\theta^\top x, \sigma^2)$, the optimal parameters are

$$q_T^*(\theta) = N(m^*, TS^*), \qquad (17)$$

where $m^*$ and $S^*$ are the solutions at $T = 1$ given in Equations (6) and (7). We prove this in Appendix A.6.

**Different Temperatures for Different Tasks.** When making predictions, there are several possible settings which would lead to different optimal temperatures. Intuitively, given the overestimation of variance in Theorem 3.7, we may expect that there is an optimal temperature $T < 1$ which will make the $T$-MFVI and exact posterior predictive distributions match most closely on in-distribution data. However, by Lemma 3.6, if there are some directions in the input space where the predictive variance is overestimated, there must also be locations where the predictive variance is underestimated. We can therefore deduce that there may be a temperature $T > 1$ which would make out-of-distribution predictions better calibrated.

Furthermore, it is not immediately obvious with which measure to compare the variational and exact posteriors. There are multiple ways to measure distances between predictive distributions, each of which captures meaningful but different notions of similarity. In our experiments in Section 5.2, we examine both ID and OOD data, and consider multiple notions of distances to the true posterior. We confirm that we should expect cold temperatures ($T < 1$) to perform well for ID data, and warmer temperatures for OOD data, and that this is generally true across multiple measures of discrepancy between distributions.

**Cold Posteriors vs. Tempered Posteriors.** There are two distinct types of sharpened posteriors: tempered posteriors, in which the prior is kept constant but the likelihood is scaled, and cold posteriors, in which both the likelihood and the prior terms are scaled (Aitchison, 2021). VI methods typically favour using a tempered approach, rather than the approach taken here, which fits naturally within the ELBO objective by simply scaling the expected log-likelihood compared to the KL term. It is not generally possible to write the cold posterior objective in such an elegant way, albeit with an exception for the Gaussian priors used here (Aitchison, 2021). In general, if the cause of the CPE is one of those in Section 2, we would advocate for the use of a tempered posterior. However, this paper is interested in how well predictions from an approximate posterior match predictions from an exact posterior, and in this case, we believe that there is a good reason to prefer the cold posterior to the tempered posterior.

Concretely, the cold posterior MFVI shares a mean with the $T = 1$ posterior in our case (Multivariate Gaussians), while the tempered posterior does not. If we applied the tempered

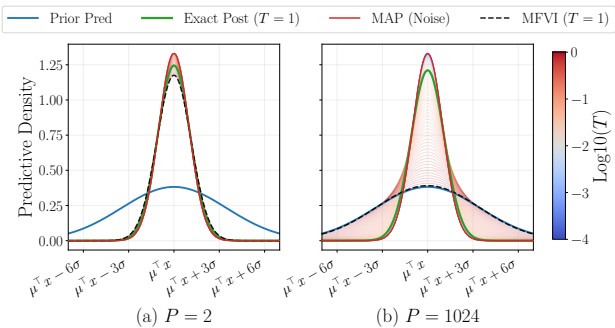

Figure 2. Posterior predictive densities at an in-distribution test point $x = \frac{1}{\sqrt{P}}\mathbf{1}_P$ for (a) $P = 2$ and (b) $P = 1024$. For $P = 2$, MFVI at $T = 1$ (dashed line) closely matches the exact posterior (green), while for $P = 1024$, MFVI at $T = 1$ is nearly indistinguishable from the prior (blue), confirming Proposition 5.1. In both cases, an appropriate cold temperature (light-coloured lines) recovers the exact posterior predictive.

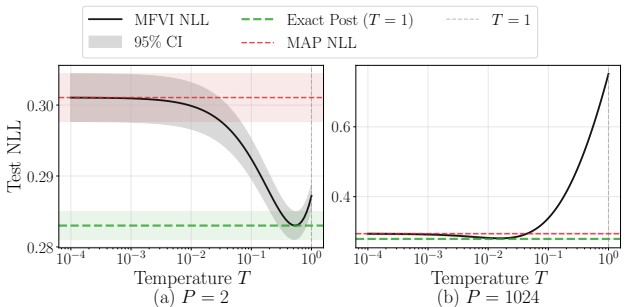

Figure 3. Test NLL as a function of temperature for ID test data, averaged over 10,000 repetitions, for (a) $P = 2$ and (b) $P = 1024$. Horizontal lines indicate the NLL of the exact posterior and the MAP solution. In both cases, an optimal temperature exists where the $T$-MFVI predictive matches the exact posterior. In high dimensions, this optimal temperature is much lower, the improvement from tuning $T$ is much larger, and the MFVI solution at $T = 1$ performs far worse than the MAP.

posterior, we would face a trade-off between the quality of our posterior predictive and the posterior mean and variance. The cold posterior avoids this trade-off by maintaining the same mean for all values of $T$.

## 5. Experiments

### 5.1. Pathological example: Low Rank Linear Regression

We now examine an extreme case that makes the pathology from Section 3 maximally severe.[1] Consider training data confined to a non-axis-aligned rank-1 subspace: $x = \varepsilon\mathbf{1}_P$ where $\varepsilon \sim \mathcal{N}(0, \beta)$. This is a direct generalization of the illustrative example from Figure 1 to arbitrary dimension $P$. The setup ensures that only a single direction of the exact

---

[1]We make our code available at https://github.com/jamesacodgers/mfvi-cpe.

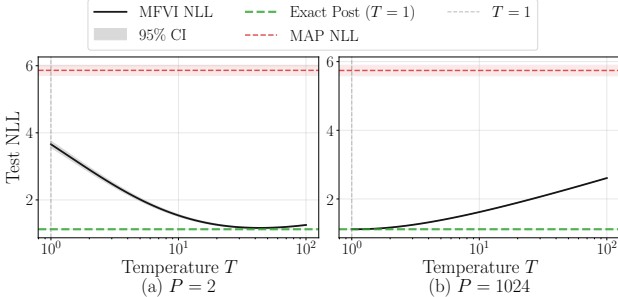

Figure 4. Test NLL as a function of temperature for OOD test data orthogonal to the training subspace, averaged over 10,000 repetitions, for (a) $P = 2$ and (b) $P = 1024$. For $P = 2$, the pattern is reversed compared to the ID case: warm posteriors ($T > 1$) improve performance. For $P = 1024$, MFVI at $T = 1$ already closely matches the exact posterior, so the optimal temperature is approximately $T \approx 1$.

posterior is informed by the data, while the remaining $P - 1$ directions retain the prior variance $\alpha^{-1}$. Figure 1 showed that MFVI overestimates predictive variance along the data direction $x_1 = x_2$; we now show that this overestimation becomes increasingly severe as the dimension grows, to the point where MFVI fails to reduce predictive variance compared to the prior at all.

**Proposition 5.1.** *In this setting, as $P \to \infty$, for any test input $x \in \mathbb{R}^P$, the posterior predictive variance of the MFVI posterior will be given by*

$$x^\top S^* x = \alpha^{-1}||x||_2^2, \tag{18}$$

*where $\alpha$ is the prior precision.*

The proof is in Appendix A.7, and the intuition is as follows: Each MFVI variance $d_p^*$ is a weighted harmonic mean of the exact posterior eigenvalues (Lemma 3.2). In our rank-1 setting, these eigenvalues consist of one small value $\delta_{\min}$ (corresponding to the data direction) and $P - 1$ values equal to the prior variance $\alpha^{-1}$. As $P$ grows, the prior eigenvalues dominate the harmonic mean, and the information from the single data direction is diluted across all $P$ axis-aligned components.

Figure 2 illustrates this effect for in-distribution test data. We evaluate at a test point $x = \frac{1}{\sqrt{P}}\mathbf{1}_P$, which lies in the data subspace and has unit norm. The figure shows posterior predictive densities for $P = 2$ (left) and $P = 1024$ (right). We highlight four key distributions: the prior predictive (which has not seen any data), the exact posterior predictive, the MAP prediction (which has zero epistemic uncertainty), and the MFVI posterior predictive at $T = 1$. For $P = 2$, the MFVI predictive closely matches the exact posterior: the approximation is working well. However, for $P = 1024$, the MFVI predictive at $T = 1$ is nearly indistinguishable from the prior, despite the exact posterior

remaining well-concentrated. This confirms the limiting behaviour of Proposition 5.1: in high dimensions, MFVI has failed to incorporate the information from the training data into its predictions.

The coloured lines show $T$-MFVI predictives for a range of temperatures. As $T \to 0$, these interpolate smoothly from the MFVI solution toward the MAP prediction. Since the exact posterior predictive lies between these two extremes, there exists a critical temperature $T^* < 1$ for which the $T$-MFVI and exact posterior predictions match exactly. Crucially, because all training data lies on a single subspace, this temperature simultaneously corrects the predictive variance for *all* in-distribution test points.

Figure 3 shows the test negative log-likelihood (NLL) as a function of temperature, averaged over 10,000 repetitions. The test set is generated ID from the same distribution as the training data, so test points also lie in the $\mathbf{1}_P$ subspace. For both $P = 2$ and $P = 1024$, lowering the temperature initially improves performance until a minimum is reached, where the $T$-MFVI predictive closely matches the exact posterior. Beyond this point, further cooling degrades performance as the predictive approaches the MAP solution.

Three key differences emerge between low and high dimensions. First, the optimal temperature for $P = 1024$ is much lower than for $P = 2$, reflecting the greater severity of the MFVI overestimation in high dimensions. Second, the potential improvement from tuning $T$ is far greater for $P = 1024$: the gap between MFVI at $T = 1$ and the exact posterior is substantial, whereas for $P = 2$ it is small. Third, the penalty for setting $T$ too low is much smaller than for setting $T$ too high. In the limit $T \to 0$, the $T$-MFVI predictive converges to the MAP solution, which has zero epistemic uncertainty, while at $T = 1$ the MFVI predictive can approach the prior (as shown in Proposition 5.1). For well-specified problems where the data is informative, the exact posterior variance lies much closer to zero than to the prior variance, so the MAP incurs a far smaller penalty than MFVI at $T = 1$.

To emphasize that no single temperature is universally optimal, we now consider predictions for test points orthogonal to the training data. Specifically, we evaluate at $x = \frac{1}{\sqrt{P}} \mathbf{1}_P^{\pm}$, where $\mathbf{1}_P^{\pm}$ is a balanced binary vector with half the entries equal to $+1$ and half equal to $-1$. This direction is orthogonal to the training subspace, so the exact posterior does not reduce predictive uncertainty compared to the prior. Figure 4 shows the test NLL for this OOD setting. For $P = 2$, the pattern from the ID case is reversed: warm posteriors ($T > 1$) now yield predictions closer to the exact posterior. This is because MFVI underestimates the predictive variance in directions orthogonal to the data, and increasing the temperature corrects this. For $P = 1024$, the picture is different: since there are so many directions orthogonal

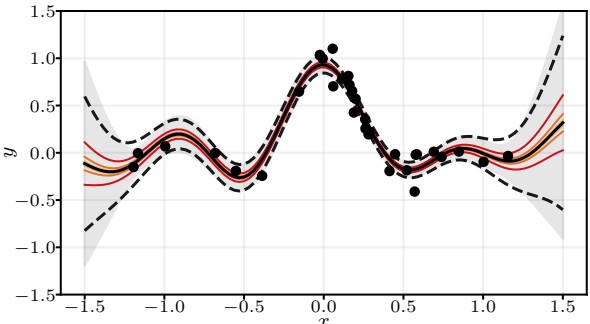

*(a)* Low dimensional feature space ($Q = 16$)

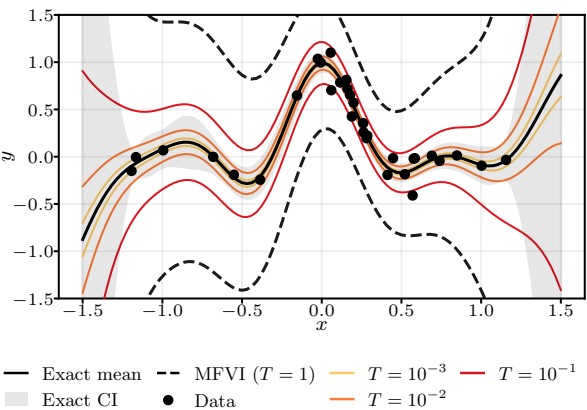

*(b)* High dimensional feature space ($Q = 1024$)

*Figure 5.* Predictions and credible intervals for fixed basis function regression with *(a)* $Q = 16$ and *(b)* $Q = 1024$ basis functions. MFVI with $T = 1$ significantly overestimates the variance of the exact posterior on in-distribution data, and lower temperatures correct this.

to the single data direction, and MFVI averages over all of them, the MFVI predictive variance in any one orthogonal direction is already close to the prior, and hence close to the exact posterior. The optimal temperature is therefore approximately $T \approx 1$. These results highlight that the optimal temperature for matching the posterior prediction is task-dependent and, for our setting at least, the CPE arises as a natural method to match the exact Bayesian prediction on in-distribution prediction tasks.

## 5.2. Basis function regression

In order to illustrate that our results from linear regression can provide a useful intuition for general Machine Learning problems, we switch our setting to Basis Function Regression. Here, we assume that we have data $\mathcal{D} = \{X, Y\}$, but we want to find relationships which would be non-linear in the original space. This can be done by defining a function $\phi : \mathbb{R}^P \to \mathbb{R}^Q$, $q = 1, \dots, Q$, and modelling the relationship between input and outputs by a weighted sum of these

functions, so

$$y_n = \theta^\top \phi(x) + \epsilon. \tag{19}$$

This problem is identical to the linear regression one introduced in Equation (1), but with inputs $\phi(x)$ rather than $x$. This type of system would typically be fitted with kernel methods, and be described as a Gaussian Process (GP) (Rasmussen and Williams, 2005). Here, we wish to see the effects of MFVI in the parameter space, so we restrict ourselves to finite values of $Q$, and calculate explicit values for $q(\theta)$, just as in the linear regression case. For our experiments, the function $\phi(\cdot)$ can either be fixed or have hyperparameters which can be learned by maximizing the marginal likelihood, as with any other GP. We use Radial Basis Functions (RBF) as our basis functions for our experiments, with the lengthscale and centroids as hyperparameters. See appendix B for more details.

**Sinc toy data.** To illustrate the underconfidence pathology we fit a $\mathrm{sinc}(x)$ function using RBF basis functions with fixed lengthscale 0.25. This lengthscale is quite high compared to the spacing of the data, meaning that the inputs in feature space, $\phi(x)$, are close together and the empirical input covariance will have very different eigenvalues. We fit the data twice: once with $Q = 16$ centroids, and once with $Q = 1024$ and show the results in Figure 5. The true posterior prediction is very similar for both of these, however, the MFVI predictions differ markedly. The MFVI prediction with $Q = 16$ matches the true posterior well, but the MFVI prediction with $Q = 1024$ is highly overinflated, matching the intuition from Section 5.1 that high-dimensional settings amplify the effect of posterior prediction mismatch.

**UCI regression.** To show how our theoretical insights translate to real-world problems, we consider several regression tasks from the UCI repository (Kelly et al., 2023), specifically the set used in Foong et al. (2019). Each data set is split into a train set, an in-distribution test set (ID Test) and an out-of-distribution test set (OOD Test). Using $Q = 500$ RBF basis functions, we train the hyperparameters as described above. We then analytically determine the mean and covariance of both the true and the $T$-MFVI posteriors (see Equations (2) and (17)) as well as the associated posterior predictive distributions. We distinguish the joint posterior predictive for a test set, $(X, Y)$, and the marginal one, for a test point, $(x, y)$. The marginal true posterior predictive at an input, $x$, is given by

$$p(y \mid x, \mathcal{D}) = \mathcal{N}(y; \mu^\top x, x^\top \Sigma x + \sigma^2), \tag{20}$$

with $\mu$ and $\Sigma$ from Equation (2). In addition to the marginal predictions we also consider the distance between the joint predictive distributions, where the covariances are $X \Sigma X^\top + \sigma^2 \mathbb{I}$ and $T X S^* X^\top + \sigma^2 \mathbb{I}$ respectively.

We are interested in establishing which temperature $T^*$ most closely matches the exact Bayesian model. However, as discussed in Section 4, it is not clear how to decide how to measure the closeness between these distributions. For this reason we consider a number of statistical divergences between the true and the $T$-MFVI posterior predictives: forward KL ($D_{KL}(p(y|x, \mathcal{D})||q_T(y|x))$), reverse KL ($D_{KL}(q_T(y|x)||p(y|x, \mathcal{D}))$), $\alpha$ divergences ($D_\alpha$, for $\alpha = 0.5$), Wasserstein-2 distance ($W_2(q_T(y|x), p(y|x, \mathcal{D}))$), and the distance between predictive variances as measured by the Frobenius norm ($\|\Sigma_p - \Sigma_{q_T}\|_F^2$). The temperature $T^*$ minimising a given divergence is recorded for 15 independent train, ID test and OOD test splits of the data, and shown in Figure 6.

A clear pattern across all divergences and all datasets can be observed, where for the training and ID test sets, all measures of divergence are minimised by $T < 1$. Meanwhile, predictions on the OOD test set always require higher temperatures than the ID test set, and, while the precise temperature will depend on the divergence used and distribution of training and OOD test points, often take values of $T > 1$. This pattern is nicely in line with our main theoretical result, Theorem 3.7, stating that the MFVI prediction will overestimate predictive uncertainty on the training data, but must underestimate predictive variance in directions with less variance in the training data to maintain the calibration criterion we give in Lemma 3.6.

Appendix B contains additional details on the divergences, data pre-processing and hyperparameter selection.

### 5.3. MFVI underconfidence in BNNs

While our paper has focused on the linear setting, the CPE is primarily of interest in Bayesian Deep Learning (BDL). A natural question is therefore whether the results we identify here also hold in BNNs. While a complete investigation of BNNs is beyond the scope of this work, there are some conceptual similarities between the setting we have studied and BDL, which we lay out here.

The key insight from our paper is that MFVI will overestimate predictive variance on ID data and underestimate predictive variance on OOD data in the linear regression setting. Both of these effects have previously been observed in MFVI-BNNs for special settings: Foong et al. (2019) showed that MFVI-BNNs' predictive variance underestimates the true posterior in certain OOD inputs, and Coker et al. (2022) showed that certain MFVI-BNNs' predictions on ID data revert to the prior as the network width increases.

Additionally, a key prediction of our theory is that the MFVI over- and underestimation of predictive variance will be worse when the regression takes place in a high-dimensional parameter space, but the data concentrates near

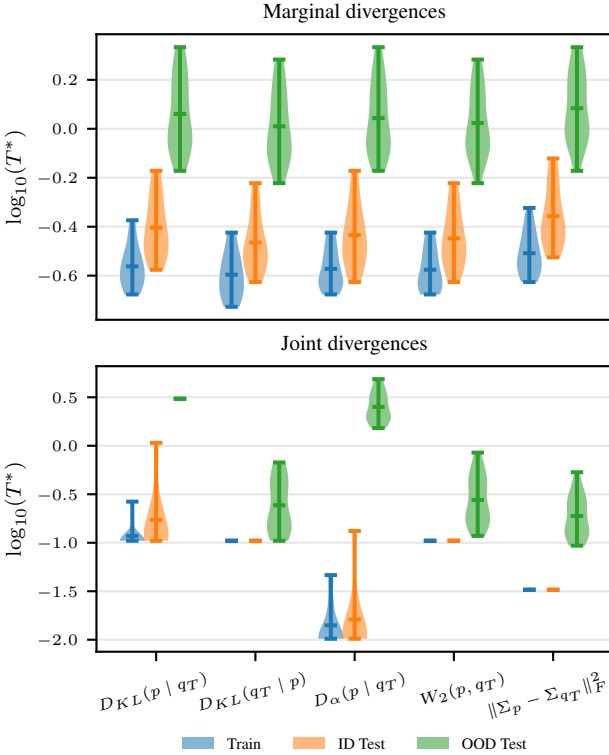

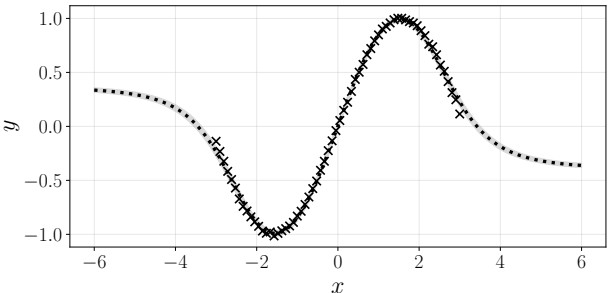

*(a)* Small BNN (depth 2, width 16)

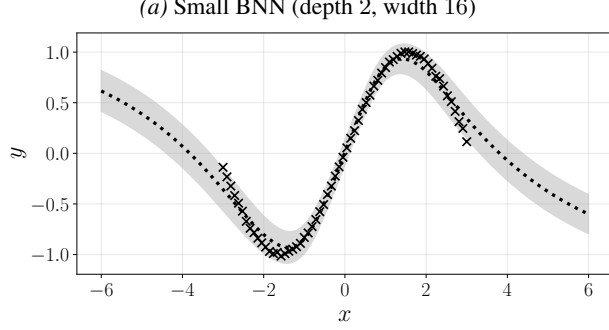

*(b)* Larger BNN (depth 2, width 512)

*Figure 7.* Plot showing the mean and credible interval for a small (Figure 7a), and large (Figure 7b) BNN trained with IVON. Similar to the basis function regression in Figure 5, as the number of parameters of the BNN increases the MFVI model becomes less confident over the region where the data was observed.

*Figure 6.* The temperature minimising a given divergence between the true posterior predictive, $p(\cdot|\cdot, \mathcal{D})$, and the $T$-MFVI predictive, $q_T(\cdot)$, on the train, ID, and OOD test sets. It can be seen that the optimal temperatures for the train and ID test set are significantly lower than for the OOD test set. For the marginal divergences, the OOD points frequently require a warm posterior ($T > 1$), while predictions at the train and ID test points benefit from a cold posterior ($T < 1$). For the joint divergences, this ordering of $T^*$ across the three sets is preserved. Results are shown on the UCI kin8nm data set with $Q = 500$ fixed RBF basis functions.

a low-dimensional subspace. It is revealing to consider the loss landscape implied by this structure. The Hessian of the log-posterior will be nearly flat in most directions, controlled only by the prior, and sharp in only the few directions where the data provides information. This is precisely the Hessian structure of the loss that has been observed in deep learning (Sagun et al., 2016).

To provide preliminary evidence that this mechanism operates beyond conjugate models, we train two BNNs on 1D sinusoidal data using IVON (Shen et al., 2024), an MFVI optimizer for deep learning. Figure 7 compares a small network (depth 2, width 16) with a larger network (depth 2, width 512). Despite the means of both networks fitting the data reasonably well, the larger network exhibits substantially wider credible intervals. This mirrors the basis function regression result in Figure 5: increasing the dimensionality of the parameter space while keeping the data fixed leads to a higher predictive variance. We believe that the overestimation of predictive variance on in-distribution

data described here may explain the success of recently developed non-mean field variational posteriors (Fadel et al., 2025).

## 6. Conclusion

In our paper, we have compared the predictions from the MFVI posterior and the exact posterior in conjugate linear regression. We have provided both theoretical and empirical evidence that, while MFVI shrinks variance in parameter space and for OOD predictions, MFVI can inflate predictive variance on ID data, which can lead to a strong CPE when data is constrained close to a low-rank linear subspace. While it is very common for approximate inference algorithms to assess predictions on both ID and OOD data (e.g., Shen et al., 2024; Fadel et al., 2025), much of the theoretical analysis of variational inference focuses on the parameter space and ignores the predictive behavior (Wainwright and Jordan, 2008; Margossian and Saul, 2023; 2025; Margossian et al., 2025; Zellinger and Vergari, 2026; Marks et al., 2026). We show that this analysis can miss important effects when the inputs are highly structured, and we hope that our work can influence other researchers to analyze the impacts of approximate Bayesian algorithms on predictions at different input locations.

## Impact Statement

This paper presents work whose goal is to advance the field of machine learning. There are many potential societal consequences of our work, none of which we feel must be specifically highlighted here.

### Acknowledgments

We thank Thomas Möllenhoff and Emtiyaz Khan for helpful discussions. VF was supported by the Branco Weiss Fellowship.

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

# A. Proofs of results

## A.1. Optimal Variational Parameters

In this section we restate and prove the optimal variational parameters.

**Lemma** (Optimal variational mean). *For any positive definite posterior covariance, the optimal mean is the mean of the posterior:*

$$m^* = \mu. \tag{21}$$

*Proof of Lemma 3.1.* If $\Sigma$ is positive definite, then $\Sigma^{-1}$ is also positive definite. By definition

$$z^\top \Sigma^{-1} z \geq 0, \tag{22}$$

with the minimum being achieved when $z = 0$. This will be achieved when $m = \mu$. $\square$

**Lemma** (Optimal variational posterior eigenvalues are harmonic means). *The optimal variational covariance, $S^*$, is given by the condition*

$$\frac{1}{d_p^*} = e_p^\top \Sigma^{-1} e_p = \sum_{q=1}^{P} \frac{w_{pq}}{\delta_q} \quad \forall\, p \in \{1, \ldots P\}, \tag{23}$$

*where $\sum_{q=1}^{P} w_{pq} = 1 \,\forall\, p \in \{1, \ldots, P\}$.*

*Proof of Lemma 3.2.* Extracting the terms which depend on $S$ from the objective, we are trying to minimise

$$F = \text{Tr}\left(\Sigma^{-1} S\right) - \log \det S. \tag{24}$$

These terms can be rewritten in terms of the eigen basis and the diagonal covariances of S as

$$\text{Tr}\left(\Sigma^{-1} S\right) = \sum_{p=1}^{P} d_p e_p^\top \Sigma^{-1} e_p. \tag{25}$$

Assuming that we restrict $d_p > 0$

$$\log \det S = \sum_{p=1}^{P} \log d_p. \tag{26}$$

Substituting these into $F$ gives

$$F = \sum_{p=1}^{P} d_p e_p^\top \Sigma^{-1} e_p - \log d_p. \tag{27}$$

Taking the derivative wrt $d_p$ and setting this equal to zero gives

$$e_p^\top \Sigma^{-1} e_p = \frac{1}{d_p^*}. \tag{28}$$

For the second equality we need to substitute the diagonalised form of the true posterior, so

$$e_p^\top \Sigma^{-1} e_p = \sum_{q=1}^{P} e_p^\top v_q v_q^\top e_p \delta_q^{-1} = \sum_{q=1}^{P} \underbrace{(e_p^\top v_q)^2}_{w_{pq}} \delta_q^{-1}. \tag{29}$$

We define $w_{pq} = (e_p^\top v_q)^2$, which must sum to one, as it is the definition of the $L_2$ norm squared of the vector $e_p$ which is defined to be equal to one. We provide a formal proof of this sum to one criterion Lemma A.1.

$\square$

**Lemma A.1.** *For any two orthonormal bases, $\{e_p\}_{p=1}^{P}$ and $\{v_q\}_{q=1}^{P}$, the weights defined by the squared inner product between the vectors, $w_{pq} = (e_p^\top v_q)^2$, obey the equality*

$$\sum_{p=1}^{P} w_{pq} = \sum_{q=1}^{P} w_{pq} = 1. \tag{30}$$

*Proof.* Consider a vector $a$ and any orthonormal basis $\{z_k\}_{k=1}^{P}$. The vector $a$ can be written as $a = \sum_{k=1}^{P}(a^\top z_k)z_k$, where the inner product $(a^\top z_k)$ defines the magnitude of the vector along each of the orthogonal directions of the basis. From this it is clear that the squared L2 distance is given, by definition, as

$$||a||_2^2 = \sum_{k=1}^{P}(a^\top z_k)^2. \tag{31}$$

Substituting $a = e_p$ and $\{v_q\}_{q=1}^{P} = \{z_k\}_{k=1}^{P}$, or $a = v_q$ and $\{e_p\}_{p=1}^{P} = \{z_k\}_{k=1}^{P}$ give the two desired equalities. $\qquad\square$

## A.2. MFVI predictions underestimate uncertainty for isotropic test points

This section gives a detailed proof that the expected predicted variance for the MFVI posterior is less than that of the exact posterior for a test point distributed as $x \sim N(0, \mathbb{I}_P)$.

One way to prove this is to first consider the predictive distribution when the test point is equal to an eigenvector of the MFVI covariance, *i.e.* at $x = e_p$ for any $p = 1, \dots, P$. At any one of these points, the variational posterior will underestimate the epistemic uncertainty of the exact posterior. Formally, we can state the following Lemma.

**Lemma A.2** (Variance Underestimation at MFVI basis vectors). *For any canonical basis vector $e_p$, $p \in \{1, \dots, P\}$ the predictive variances are governed by*

$$e_p^\top \Sigma e_p \geq e_p^\top S^* e_p. \tag{32}$$

*Proof of Lemma A.2.* For the true posterior, the predictive variance is given by

$$e_p^\top \Sigma e_p = \sum_{q=1}^{P} w_{pq} \delta_q, \tag{33}$$

which is the Arithmetic Mean (AM) of the eigenvalues of the true posterior with weights $\{w_{pq}\}_{q=1}^{P}$.

For the MFVI posterior, the predictive variance is given by

$$e_p^\top S^* e_p = d_p = \frac{1}{\sum_{q=1}^{P} w_{pq} \delta_q^{-1}}, \tag{34}$$

which is the Harmonic Mean (HM) of the eigenvalues of the true posterior with the same weights. The well known AM-HM inequality gives the desired result. $\qquad\square$

By noting that the quadratic form $e_p^\top A e_p = A_{pp}$, with $A_{pp}$ being the $p$th diagonal element of the matrix $A$, the result above also easily extends to the following inequality for the trace.

**Lemma A.3** (Trace Inequality.). *The trace of the MFVI posterior and the true posterior are governed by*

$$Tr(\Sigma) \geq Tr(S^*). \tag{35}$$

*Proof of Lemma A.3.* In the basis of the MFVI posterior, trace is given by

$$\mathrm{Tr}(\Sigma) = \sum_{p=1}^{P} e_p^\top \Sigma e_p, \quad \mathrm{Tr}(S^*) = \sum_{p=1}^{P} e_p^\top S^* e_p. \tag{36}$$

Each of these terms $p = 1, \dots, P$ is governed by the inequality in Lemma A.2, meaning each term in this sum can be bound by $e_p^\top \Sigma e_p \geq e_p^\top S^* e_p$, hence the sum of all these terms must also be bound. $\qquad\square$

From here, we can directly move to proving the isotropic variance overestimation result.

**Lemma.** *For a test point $x \sim N(0, \mathbb{I}_P)$, the predictive variances of the MFVI posterior and exact posterior are governed by*

$$\mathbb{E}_{x \sim N(0, \mathbb{I}_P)} \left[ x^\top \Sigma x \right] \geq \mathbb{E}_{x \sim N(0, \mathbb{I}_P)} \left[ x^\top S^* x \right] \tag{37}$$

*Proof.* By noting

$$\mathbb{E}_{x \sim N(0, \mathbb{I}_P)} \left[ x^\top \Sigma x \right] = \mathrm{Tr}\left( \Sigma \right), \tag{38}$$

$$\mathbb{E}_{x \sim N(0, \mathbb{I}_P)} \left[ x^\top S^* x \right] = \mathrm{Tr}\left( S^* \right), \tag{39}$$

the result follows directly from A.3. $\qquad \square$

## A.3. MFVI overestimates uncertainty along the first principal component of the training data

Here we provide the proof of Lemma 3.4, with the argument relating this result to the distribution of the training inputs being given in the main text.

**Lemma.** *(Overestimation of predictive variance in one direction of the input space.) For points in the input space defined by $\tilde{x} \in span(v_{q^*})$ for some scalar, the predictive variance for the exact posterior and the MFVI posterior are governed by the inequality*

$$\tilde{x}^\top S^* \tilde{x} \geq \tilde{x}^\top \Sigma \tilde{x}. \tag{40}$$

*Proof of Lemma 3.4.* Let $\tilde{x} = c v_{q^*}$, for some constant $c$. For $c = 0$ the equality holds trivially. For any non-zero constant $c$, the predictive variance of the two posteriors are given by

$$\tilde{x}^\top S^* \tilde{x} = c^2 v_{q^*}^\top S v_{q^*} \quad \text{and} \quad \tilde{x}^\top \Sigma \tilde{x} = c^2 v_{q^*}^\top \Sigma v_{q^*}, \tag{41}$$

so proving the result for $c = 1$ gives the result for all other non zero values of $c$.

The predictive variance of the MFVI posterior is given by

$$v_{q^*}^\top S^* v_{q^*} = v_{q^*}^\top \left( \sum_{p=1}^{P} e_p e_p^\top d_p \right) v_{q^*} = \sum_{p=1}^{P} (v_{q^*}^\top e_p)^2 d_p. \tag{42}$$

Note that $(v_{q^*}^\top e_p)^2 = w_{pq^*}$ which is the weighted term relating the MFVI posterior eigenvalues to the exact posterior eigenvalues. Making this substitution, along with the values of the eigenvalues of $d_p$ from Lemma 3.2, we can use

$$v_{q^*}^\top S^* v_{q^*} = \sum_{p=1}^{P} \frac{w_{pq^*}}{\sum_{q=1}^{P} w_{pq} \delta_q^{-1}}. \tag{43}$$

As we have defined $\delta_{q^*}$ to be the minimum eigenvalue, we can use the inequality

$$v_{q^*}^\top S^* v_{q^*} \geq \sum_{p=1}^{P} \frac{w_{pq^*}}{\sum_{q=1}^{P} w_{pq} \delta_{q^*}^{-1}} = \sum_{p=1}^{P} \frac{w_{pq^*}}{\delta_{q^*}^{-1}} = \sum_{p=1}^{P} w_{pq^*} \delta_{q^*} = \delta_{q^*} = v_{q^*} \Sigma v_{q^*}. \tag{44}$$

This proves the statement for $c = 1$, hence giving the result for all non-zero c.

$\qquad \square$

Following this we can simply show the overestimation of predictive variance along the first principal component.

**Theorem.** *Consider the conjugate Bayesian linear model of Equation (1) with a spherical prior $\theta \sim N(0, \alpha^{-1} \mathbb{I})$, and let $w$ denote the first principal component of the training inputs $X$. Then for any test point $\tilde{x} \in span(w)$, the predictive variances of the MFVI and exact posteriors satisfy*

$$\tilde{x}^\top S^* \tilde{x} \geq \tilde{x}^\top \Sigma \tilde{x}. \tag{45}$$

*Proof of Theorem 3.5.* For a spherical prior $\theta \sim N(0, \alpha^{-1}\mathbb{I})$, the exact posterior covariance $\Sigma$ shares its eigenbasis with the Gram matrix $X^\top X$, since $\alpha\mathbb{I}$ commutes with every matrix. Writing this shared eigenbasis as $\{v_q\}_{q=1}^P$ with associated Gram matrix eigenvalues $\{\gamma_q\}_{q=1}^P$, the posterior eigenvalues are

$$\delta_q = \left(\alpha + \frac{\gamma_q}{\sigma^2}\right)^{-1}. \tag{46}$$

This relationship is strictly decreasing in $\gamma_q$, so the eigenvector associated with the largest Gram matrix eigenvalue is the eigenvector associated with the smallest posterior eigenvalue. The former is, by definition, the first principal component of the training data, $w$, and the latter is $v_{q^*}$ as defined in Lemma 3.4, so $w = v_{q^*}$.

Hence $\mathrm{span}(w) = \mathrm{span}(v_{q^*})$, and any test point $\tilde{x} \in \mathrm{span}(w)$ satisfies the hypothesis of Lemma 3.4. Applying that lemma directly gives

$$\tilde{x}^\top S^* \tilde{x} \geq \tilde{x}^\top \Sigma \tilde{x}, \tag{47}$$

as claimed. $\qquad\square$

### A.4. MFVI under- and overestimates uncertainties similarly

The easiest way to show Lemma 3.6 is to first show a particular quantity, $\mathrm{Tr}\left(\Sigma^{-1}S^*\right)$, is conserved for any exact posterior covariance.

**Lemma A.4** (Conserved Trace Product). *For any eigenbasis of the MFVI posterior, the optimal MFVI posterior covariance will obey*

$$Tr\left(\Sigma^{-1}S^*\right) = P. \tag{48}$$

*Proof.* The trace of any $P \times P$ matrix $A$ can be calculated by the sum of quadratic forms of the orthonormal eigenbasis of the MFVI posterior: $\mathrm{Tr}\,(A) = \sum_{p=1}^P e_p^\top A e_p$. Applying this to the product of the inverse of the true posterior covariance matrix with the MFVI posterior covariance matrix gives

$$\mathrm{Tr}\left(\Sigma^{-1}S^*\right) = \sum_{p=1}^P e_p^\top \Sigma^{-1}S^* e_p = \sum_{p=1}^P d_p^* \left(e_p^\top \Sigma^{-1} e_p\right) = \sum_{p=1}^P 1 = P. \tag{49}$$

$\qquad\square$

From this it is easy to show Lemma 3.6.

**Lemma** (Calibrated MFVI). *Consider the set of eigenvectors of the true posterior $\{v_q\}_{q=1}^P$. For these points*

$$\frac{1}{P} \sum_{q=1}^P R(v_q) = 1. \tag{50}$$

*Proof of Lemma 3.6.* For any matrix $A$ and orthonormal basis $\{v_q\}_{q=1}^P$, we have $\mathrm{Tr}(A) = \sum_{q=1}^P v_q^\top A v_q$. Applying this to $A = \Sigma^{-1}S^*$ gives

$$\mathrm{Tr}(\Sigma^{-1}S^*) = \sum_{q=1}^P v_q^\top \Sigma^{-1}S^* v_q. \tag{51}$$

Since $v_q$ is an eigenvector of $\Sigma$ with eigenvalue $\delta_q$, we have $v_q^\top \Sigma^{-1} = \frac{1}{\delta_q} v_q^\top$. Thus

$$\mathrm{Tr}(\Sigma^{-1}S^*) = \sum_{q=1}^P \frac{v_q^\top S^* v_q}{\delta_q} = \sum_{q=1}^P \frac{v_q^\top S^* v_q}{v_q^\top \Sigma v_q} = \sum_{q=1}^P R(v_q). \tag{52}$$

From Lemma A.4 this equals $P$, and dividing both sides by $P$ gives the result. $\qquad\square$

## A.5. MFVI overestimates predictive variance for data with the empirical covariance

**Theorem.** *For test data points distributed according to $x \sim \hat{p}(x)$, the difference in the expected predicted variance is given by*

$$\mathbb{E}_{x \sim \hat{p}(x)} \left[ x^\top \Sigma x - x^\top S^* x \right] \leq 0. \tag{53}$$

*Proof of Theorem 3.7.* Due to the linearity of the expectation, we can write

$$\mathbb{E}_{x \sim \hat{p}(x)} \left[ x^\top \Sigma x - x^\top S^* x \right] = \mathbb{E}_{x \sim \hat{p}(x)} \left[ x^\top \Sigma x \right] - \mathbb{E}_{x \sim \hat{p}(x)} \left[ x^\top S^* x \right], \tag{54}$$

which allows us to deal with the two expectations separately.

To prove the result we need to make use of the following identity:

$$\hat{\Gamma} = \frac{\sigma^2}{N} \left( \Sigma^{-1} - \alpha \mathbb{I} \right). \tag{55}$$

By making use of the identity in Equation (55), we can write

$$
\begin{aligned}
\mathbb{E}_{x \sim \hat{p}(x)} \left[ x^\top \Sigma x \right] &= \mathrm{Tr} \left( \hat{\Gamma} \Sigma \right) \\
&= \mathrm{Tr} \left( \frac{\sigma^2}{N} \left( \Sigma^{-1} - \alpha \mathbb{I} \right) \Sigma \right) \\
&= \frac{\sigma^2}{N} \left( \mathrm{Tr} \left( \Sigma^{-1} \Sigma \right) - \alpha \mathrm{Tr} \left( \Sigma \right) \right) \\
&= \frac{\sigma^2}{N} \left( \mathrm{Tr} \left( \mathbb{I} \right) - \alpha \mathrm{Tr} \left( \Sigma \right) \right) \\
&= \frac{\sigma^2}{N} \left( P - \alpha \mathrm{Tr} \left( \Sigma \right) \right).
\end{aligned} \tag{56}
$$

We can make use of Lemma A.4 to see

$$
\begin{aligned}
\mathbb{E}_{x \sim \hat{p}(x)} \left[ x^\top S^* x \right] &= \mathrm{Tr} \left( \hat{\Gamma} S^* \right) \\
&= \mathrm{Tr} \left( \frac{\sigma^2}{N} \left( \Sigma^{-1} - \alpha \mathbb{I} \right) S^* \right) \\
&= \frac{\sigma^2}{N} \left( \mathrm{Tr} \left( \Sigma^{-1} S^* \right) - \alpha \mathrm{Tr} \left( S^* \right) \right) \\
&= \frac{\sigma^2}{N} \left( P - \alpha \mathrm{Tr} \left( S^* \right) \right).
\end{aligned} \tag{57}
$$

Subtracting one of these from the other gives

$$\mathbb{E}_{x \sim \hat{p}(x)} \left[ x^\top \Sigma x \right] - \mathbb{E}_{x \sim \hat{p}(x)} \left[ x^\top S^* x \right] = \frac{\sigma^2 \alpha}{N} \left( \mathrm{Tr} \left( S^* \right) - \mathrm{Tr} \left( \Sigma \right) \right). \tag{58}$$

As $\frac{\sigma^2 \alpha}{N} > 0$, this value will take the same sign as the difference in the traces. From Lemma A.3 we know that

$$\mathrm{Tr} \left( S^* \right) - \mathrm{Tr} \left( \Sigma \right) \leq 0, \tag{59}$$

hence the expected posterior predictive variance of the true posterior is less than the expected posterior predictive variance of the MFVI posterior. □

## A.6. Cold Posteriors

**MFVI posterior rescaling for cold posterior.** Here, we prove that the cold posterior MFVI objective has the form $q_T(\theta) = N(m^*, T S^*)$ as stated in Section 4, where $m^*$ and $S^*$ are the solutions at $T = 1$ given in Equations (6) and (7).

*Proof.* In the conjugate case with prior $N(0, \alpha^{-1}\mathbb{I}_P)$ and likelihood $N(y|\theta^\top x, \sigma^2)$, the cold posterior is given by

$$p_T(\theta|\mathcal{D}) \propto \left[\prod_{n=1}^{N} N(y_n|\theta^\top x_n, \sigma^2)^{\frac{1}{T}}\right] N(\theta|0, \alpha^{-1}\mathbb{I})^{\frac{1}{T}} \tag{60}$$

$$\propto \left[\prod_{n=1}^{N} \exp\left(-\frac{1}{2\sigma^2}\left(y_n - \theta^\top x_n\right)^2\right)^{\frac{1}{T}}\right] \exp\left(-\frac{\alpha}{2}\theta^2\right)^{\frac{1}{T}} \tag{61}$$

$$= \left[\prod_{n=1}^{N} \exp\left(-\frac{1}{2T\sigma^2}\left(y_n - \theta^\top x_n\right)^2\right)\right] \exp\left(-\frac{\alpha}{2T}\theta^2\right) \tag{62}$$

$$\propto \left[\prod_{n=1}^{N} N(y_n|\theta^\top x_n, T\sigma^2)\right] N\left(\theta|0, \frac{T}{\alpha}\mathbb{I}_P\right). \tag{63}$$

In other words, this is the exact posterior for a system with a rescaled likelihood and prior.

Applying the known results for the covariance gives

$$\begin{aligned}
\Sigma_T &= \left(\frac{\alpha}{T}\mathbb{I}_P + \frac{1}{T\sigma^2}X^\top X\right)^{-1} \\
&= T\left(\alpha\mathbb{I}_P + \frac{1}{\sigma^2}X^\top X\right)^{-1} \\
&= T\Sigma,
\end{aligned} \tag{64}$$

so the exact cold-posterior covariance is the exact posterior at $T = 1$ scaled by a factor T.

Applying the known result for the mean gives

$$\mu_T = \Sigma_T\left(\frac{1}{T\sigma^2}X^\top Y\right) = \frac{1}{\sigma^2}\Sigma X^\top Y = m, \tag{65}$$

so applying the exponentiation does not change the mean of the posterior.

From here it is straightforward to apply the optimal variational parameter arguments to get

$$m_T = m = \mu \; \forall \; T, \tag{66}$$

so the change in temperature does not effect the mean values.

Similarly,

$$d_{pT} = \frac{1}{e_p^\top \frac{1}{T}\Sigma^{-1}e_p} = \frac{T}{e_p^\top \Sigma^{-1}e_p} = Td_p, \tag{67}$$

so

$$S_T = TS^*. \tag{68}$$

$\square$

## A.7. Pathological example: Low Rank Linear Regression

**Proposition.** *As $P \to \infty$, for any test input $x \in \mathbb{R}^P$ the posterior predictive variance of the MFVI posterior will be given by*

$$x^\top S^* x = \alpha^{-1}||x||_2^2, \tag{69}$$

*where $\alpha$ is the prior precision.*

*Proof of Proposition 5.1.* We start this proof by considering the symmetries, thereby giving us the weights which relate the eigenvalues of the exact posterior to the MFVI posterior. As the relation between $1_P$ and each canonical basis function

is identical, so it is clear that all values of $w_{pq}$ are the same. Combining this with the sum-to-one requirement gives $w_{pq} = \frac{1}{P} \forall p, q$.

Next we note that in the exact posterior all of the eigenvalues will be $\alpha^{-1}$, apart from the one associated with the eigenvector which points towards the span of the data, which we will denote by $\delta_{min}$.

We can then consider the predicted variance which would be given by the normalised test point $\frac{x}{||x||_2^2}$ this into the predicted variance gives

$$\frac{x^\top S^* x}{||x||_2^2} = \sum_{p=1}^{P} \frac{w_{pq}}{\sum_{q=1}^{P} \frac{w_{pq}}{\delta_q}} \tag{70}$$

$$= \sum_{p=1}^{P} \frac{\frac{1}{P}}{\sum_{q=1}^{P} \frac{1}{P\delta_q}} \tag{71}$$

$$= \sum_{p=1}^{P} \frac{\frac{1}{P}}{\frac{1}{P\delta_{min}} + (P-1)\alpha} \tag{72}$$

$$= \frac{1}{\frac{1}{P\delta_{min}} + \frac{(P-1)\alpha}{P}} \tag{73}$$

Inspecting the denominator

$$\frac{1}{P\delta_{min}} + \frac{(P-1)\alpha}{P} = \frac{1}{P\delta_{min}} - \frac{\alpha}{P} + \alpha. \tag{74}$$

As $P \to \infty$ the first two terms go to zero, leaving $\alpha$. Applying the limit quotient rule we see

$$\frac{x^\top S^* x}{||x||_2^2} = \alpha^{-1} \tag{75}$$

and hence

$$x^\top S^* x = \alpha^{-1} ||x||_2^2. \tag{76}$$

$\square$

# B. Experiment Details

**Pre-processing and Train-Test Split.** Given a full data set $D = (X, Y)$ as defined in Section 3, we make a split into a train, ID test and OOD test set as follows. First, the OOD test set is formed by sorting and then splitting the data on a chosen feature. The remainder is randomly split into a train and an ID test set. All inputs are then standardized using the train set mean and standard deviation, and the outputs are all centred with the train set output mean. The random splits are independent across trials, giving the desired variability in train-test splits, as well as learned hyperparameters (see below).

**Basis functions.** All experiments use RBF basis functions to construct the feature map, $\varphi : \mathbb{R}^P \mapsto \mathbb{R}^Q$, with

$$\varphi(x) = \left(e^{-\frac{||x-c_q||_2^2}{2l}}\right)_{q=1}^{Q}, \tag{77}$$

for basis centroids $\{c_q\}_{q=1}^{Q}$ where $c_q \in \mathbb{R}^P$ and a common lengthscale, $l$. The centroids are sampled independently as

$$c_q \sim \mathcal{N}(0_P, M), \tag{78}$$

with $M = \frac{1}{n_{train}} X_{train}^\top X_{train}$, the empirical covariance of the standardized training inputs. We denote the inputs, $X$, mapped to this basis by $\Phi_l$, where the subscript makes explicit the dependence on the learnable lengthscale.

**Learning hyperparameters.** The three learnable hyperparameters in our experiments are the observation noise, $\sigma$, the prior precision, $\alpha$, and the RBF lengthscale, $l$. They are chosen by gradient-based optimization of the train set marginal likelihood, given by

$$p(Y_{train} \mid X_{train}) = \mathcal{N}(0_{n_{train}}, \alpha^{-1}\Phi_{l,train}\Phi_{l,train}^\top + \sigma^2 I_{n_{train}}), \tag{79}$$

and, for numerical stability, we minimise the negative log likelihood,

$$\mathcal{L}(\sigma, \alpha, l) = -\log p(Y_{train} \mid X_{train}). \tag{80}$$

To this end, we use the ADAM optimizer (Kingma and Ba, 2015) with default settings for a fixed 2000 gradient steps.

**Posterior predictive.** Both the true and $T$-MFVI posterior predictive considered in our experiments are analytically tractable in the conjugate BLR setting. We distinguish the joint posterior predictive for a test set, $(X, Y)$ and the marginal one, for a test point, $(x, y)$. The marginal true posterior predictive at an input, $x$, is given by

$$p(y \mid x, \mathcal{D}) = \int p(y \mid \theta, x)\, p(\theta \mid \mathcal{D})\, d\theta \tag{81}$$

$$= \mathcal{N}(y; \mu^\top x, x^\top \Sigma x + \sigma^2), \tag{82}$$

with $\mu$ and $\Sigma$ from Equation (2). The marginal $T$-MFVI posterior predictive is

$$q_T(y \mid x, \mathcal{D}) = \int p(y \mid \theta, x)\, q_T(\theta \mid \mathcal{D})\, d\theta \tag{83}$$

$$= \mathcal{N}(y; m^{*\top} x, T x^\top S^* x + \sigma^2), \tag{84}$$

with $m^*$ and $S^*$ from Equations (6) and (7). For the joint posterior predictives, we have

$$p(Y \mid X, \mathcal{D}) = \int p(Y \mid \theta, X)\, p(\theta \mid \mathcal{D})\, d\theta \tag{85}$$

$$= \mathcal{N}(Y; X\mu, X\Sigma X^\top + \sigma^2 I), \tag{86}$$

and

$$q_T(Y \mid X, \mathcal{D}) = \int p(Y \mid \theta, X)\, q_T(\theta \mid \mathcal{D})\, d\theta \tag{87}$$

$$= \mathcal{N}(y; Xm^*, TXS^*X^\top + \sigma^2 I). \tag{88}$$

**Divergences.** A number of divergences are considered in Figure 6, with the aim of quantifying how close the $T$-MFVI posterior predictive is to the true posterior predictive. While we write the divergences for the marginal posterior predictives here, they straightforwardly extend to the joint predictive setting. We use both the forward and reverse Kullback-Leibler (KL) divergence,

$$D_{KL}(p\|q_T) = \int p(y, \mid x, \mathcal{D}) \log \frac{p(y, \mid x, \mathcal{D})}{q_T(y, \mid x, \mathcal{D})}\, dy, \tag{89}$$

$$D_{KL}(q_T\|p) = \int q_T(y, \mid x, \mathcal{D}) \log \frac{q_T(y, \mid x, \mathcal{D})}{p(y, \mid x, \mathcal{D})}\, dy. \tag{90}$$

$$\tag{91}$$

We also consider the standard $\alpha$-divergence at $\alpha = 0.5$,

$$D_\alpha(p, q_T) = 4\left(1 - \int \sqrt{p(y, \mid x, D)q_T(y, \mid x, D)}\, dy\right), \tag{92}$$

making it proportional to Hellingers distance, and thus symmetric. Further, we make use of the 2-Wasserstein distance, which is analytic for two Gaussians and given by

$$W_2(p, q_T) = \inf_{\gamma \in \Gamma(p, q_T)} \mathbb{E}_{(y, y') \sim \gamma}[\|y - y'\|^2]^{\frac{1}{2}}, \tag{93}$$

with $\Gamma(p, q_T)$ the set of all couplings of the two posterior predictive distributions. Lastly, for comparing marginal predictives, we consider the simple squared difference between predictive variances

$$(\sigma_p^2 - \sigma_{q_T}^2)^2 \equiv (x^\top S^* x + \sigma^2 - x^\top \Sigma x - \sigma^2)^2 \tag{94}$$
$$= (x^\top S^* x - x^\top \Sigma x)^2, \tag{95}$$

which we generalize to the squared Frobenius norm

$$\|\Sigma_p - \Sigma_{q_T}\|_F^2 \equiv \|X\Sigma X^\top + \sigma^2 I - TXS^* X^\top - \sigma^2 I\|_F^2 \tag{96}$$
$$= \|X\Sigma X^\top - TXS^* X^\top\|_F^2, \tag{97}$$

for comparing joint predictives.

When considering the marginal posterior predictives for a given divergence, the marginal divergences are simply averaged across test points, reflecting a marginal per-data-point predictive divergence. For the joint posterior predictves, each test set simply gives one divergence between the two multivariate Gaussians. This is done for a grid of 100 values $T$ (equally spaced on the log scale), and $T^*$ in Figure 6 is then found as the divergence minimiser on this grid.

## C. Additional Experiment Results

Here we present experimental results following the set up of Appendix B and Section 5.2, for additional UCI data sets. Analogous to Figure 6, Figures 8 and 9 show the temperature minimising various measures of divergence between the true posterior predictive and the $T-$MFVI posterior predictive.

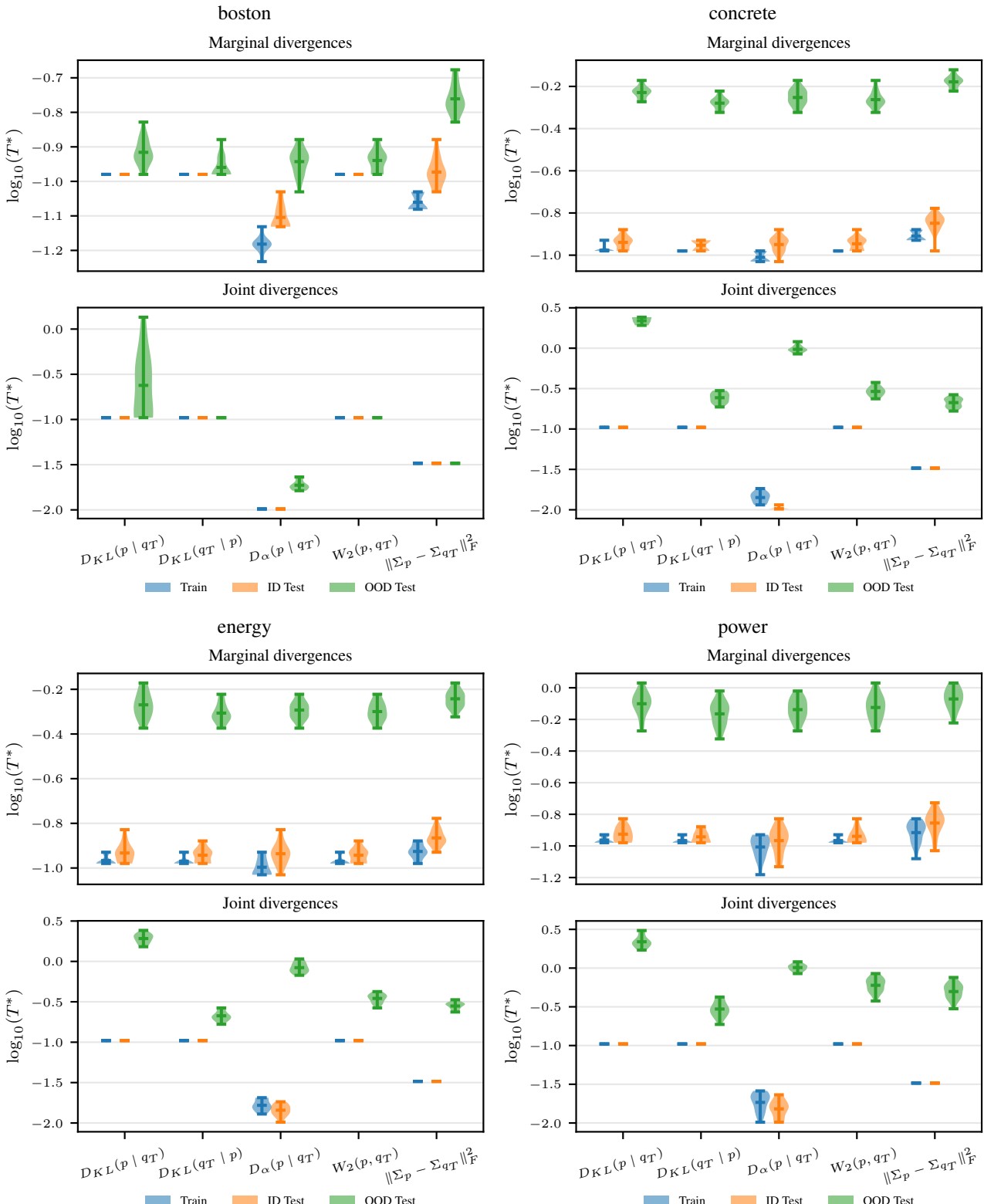

*Figure 8.* Optimal temperatures for various measures of divergence are shown across independent train-test splits.

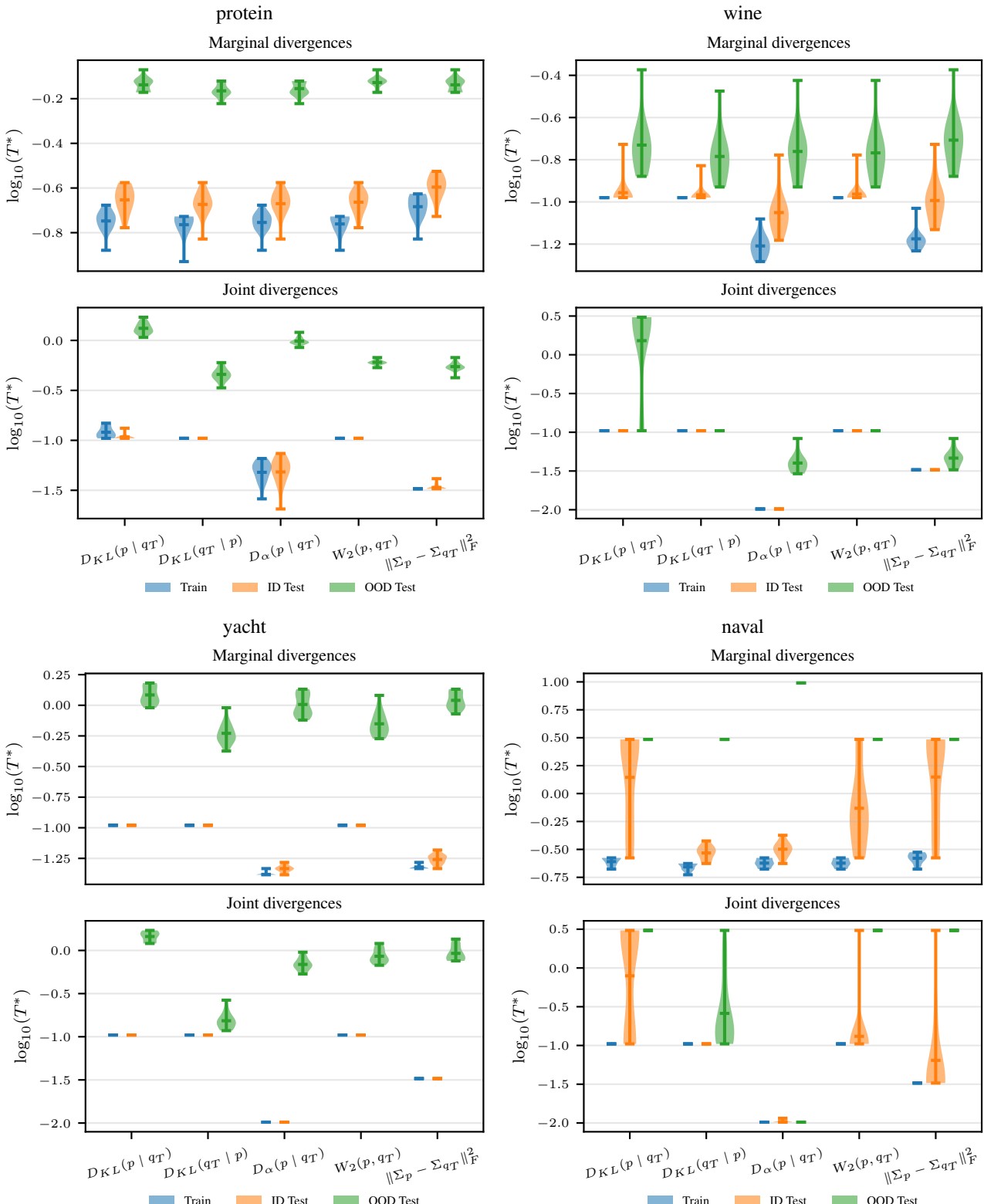

*Figure 9.* Optimal temperatures for various measures of divergence are shown across independent train-test splits.

