# OpenReview forum: "Gaussian Mean Field Variational Inference can Overestimate Predictive Variance"
_ICML.cc/2026/Conference — ICML 2026 regular_

### Official Review · Reviewer_KAMs · 2026-03-01

**Soundness:** 3
**Presentation:** 3
**Significance:** 2
**Originality:** 3
**Overall Recommendation:** 4
**Confidence:** 3

**Summary:**

The paper takes a close look at Gaussian mean-field variational inference in a setting where the exact Bayesian answer is known (Bayesian linear regression). In parameter space, the diagonal covariance does tend to miss correlations and shrink uncertainty in many directions, but when you move to what people actually care about in practice (predictive uncertainty at test inputs), the story can flip. For test points that look like the training data, MFVI can end up giving larger predictive variance than the exact posterior. The authors also give an example where this mismatch becomes especially stark in high dimensions. As a practical angle, they discuss temperature scaling (cold posteriors) as a way to dial predictive variance up or down.

**Compliance With Llm Reviewing Policy:**

Affirmed.

**Final Justification:**

While the current results are certainly limited to the conjugate linear setting, they nonetheless provide valuable insights and a solid starting point for deeper investigation of the non-conjugate setting. Furthermore, the authors have provided thorough and detailed explanations that adequately address my concerns, and their response has convinced me of the merits of this work. I therefore recommend acceptance.

**Key Questions For Authors:**

### Questions for the Authors

1. **Temperature selection:** Building on the paper’s insights and the discussion of cold MFVI, do you have principled guidance for how practitioners should choose the temperature parameter in practice?

2. **Beyond conjugate linear regression:** Do you have theoretical arguments or empirical evidence that the in-distribution predictive overestimation effect persists in nonconjugate settings?

3. **Robustness to shift:** How sensitive is the predictive overestimation phenomenon to moderate covariate shift, that is, when the test input distribution differs from the training distribution?

**Strengths And Weaknesses:**

### Strengths

* **Sharp and surprising message:** It provides a rigorous analysis of broad folk belief (“MFVI tends underestimates uncertainty”) by showing that predictive variance can go the other way, especially on in-distribution inputs.
* **Clean theory in a tractable model:** Working in conjugate Bayesian linear regression makes the mechanisms transparent and the claims provable rather than anecdotal.
* **Geometry-based insight:** The paper clarifies why this happens (diagonal covariance + misalignment with data/posterior directions), which is useful beyond the specific model.

### Weaknesses

* **Limited practical guidance beyond temperature tuning:** While the paper’s message is clear, it falls short of translating the insight into a broadly usable recipe. Tempering is presented as the primary remedy, yet the paper does not offer a principled, generally applicable way to select the temperature, nor does it provide alternative diagnostics or correction strategies when the exact posterior predictive is unavailable.
* **Relies heavily on a special setting:** The clean scaling and interpretation depend on conjugacy, and it remains unclear how much of the phenomenon carries over to nonconjugate models or deep networks where optimization and approximation errors are intertwined.
* **Empirical coverage could be broader:** The experiments support the theory in the linear Gaussian regime, but the paper would be stronger with additional examples that meaningfully go beyond this setting and that demonstrate the effect, and any proposed remedies, under more realistic model misspecification and nonlinear prediction.

---

> ### Author Rebuttal · Authors · 2026-03-30
>
> We thank the reviewer for their review, and provide responses to questions and concerns below.
>
> > Q1. & W1. Temperature selection.
>
> We agree that further research on how to address the overestimation of predictive variance is needed, and we plan to address this more thoroughly in future work. However, we are not certain that any method for temperature selection would represent a generally applicable method, and suspect that better methods for correcting predictive variance overestimation will need to be developed to be generally applicable.
>
> In Section 4, under the subheading **Different Temperatures for Different Tasks**, we argue that there is no single best temperature for all tasks. The reason for this is that, in high dimensions, the MFVI posterior contains significantly less information than the exact posterior. Adding the single scalar of $T$ cannot contain all the information which is lost by removing the diagonal elements of the posterior covariance matrix, so it is necessary to select which features of the posterior are important for the particular task and tune $T$ to preserve those. The particular example of this which we focused on was whether to preserve predictive distributions for in-distribution or out-of-distribution points, which we theoretically and empirically justify will require different temperatures.
>
> The goal of linking the CPE to MFVI posterior variance overestimation is to improve the theoretical justification for the existing practice of setting $T<1$ for in-distribution data, rather than to offer a novel solution. Practically, we would recommend selecting temperatures with the same method as selecting any other hyper-parameter: by the performance on a validation set.
>
> We will happily include the above points in the next version of our paper.
>
> > Q2. Beyond conjugate linear regression, W2. Relies heavily on a special setting, W3. Empirical coverage could be broader
>
> Firstly, we would like to state that we believe that the linear regression case is important and the surprisingly strong results in this case will be of interest to the community. However, in response to this and other comments, we have decided to include a small deep learning example. We have included the full text in our response to Reviewer ELpz, but a brief summary is here.
>
> The theoretical justification for believing our results are relevant to deep leaning is that the examples in which we demonstrate MFVI severely overestimates the predictive variance share a similar geometry to the loss-landscape of deep neural networks. Specifically, the examples where we show that linear regression significantly overestimates predictive variance have a high curvature of the log-posterior density in only a few directions and low curvature in a much greater number. This Hessian structure is exactly the one which is present in deep learning [1].
>
> We will include a demonstration of a similar behaviour to the basis function regression in a new [Figure 7](https://anonymous.4open.science/r/anonymous_fig-171F/fig7.png). In it we show that, similar to Figure 5, as the number of parameters of the model is increased, the predictions at the data locations become less confident for MFVI.
>
> We hope that including this additional discussion and result will convince the reviewer that our research is relevant to deep learning.
>
> [1] Sagun, Levent, Leon Bottou, and Yann LeCun. "Eigenvalues of the hessian in deep learning: Singularity and beyond." https://arxiv.org/abs/1611.07476
>
> > Q3. Robustness to shift
>
> The overestimation is specific to in-distribution predictions, and for out-of-distribution data the reverse occurs, with MFVI underestimating predictive variance. We make this point at several points in the paper, though we will attempt to make this point more prominent.
>
> This is visible in Figure 1(a), where the red (overestimation) region aligns with the training data, but the blue (underestimation) region covers most of the input space. We then make this more rigorous in Lemma 3.3, which shows that variance underestimation will occur for isotropic test distributions, and Lemma 3.5, which shows that overestimation along data-aligned directions necessarily implies variance underestimation elsewhere.
>
> We then demonstrate that for out-of-distribution points a warm posterior effect occurs, *i.e.* predictions match the exact Bayesian predictions by increasing the predictive variance.  We confirm this behaviour experimentally in Figures 4 and 6.
>
> In the camera-ready version we will make this point more explicit, particularly in the discussion of Figure 1(a) and after Lemma 3.3.

---

> > ### Author Rebuttal · Reviewer_KAMs · 2026-04-02
> >
> > Sorry for the confusion in my third question, and addressing the other questions 1 and 2. To clarify, my third question was intended in the context of non-conjugate models rather than the linear case. With that in mind, for non-conjugate models (e.g., neural networks with non-Gaussian likelihoods), is it still possible to recover an analogous or similar observation of the overestimation/underestimation transition under covariate shift? For instance, does the geometric structure of the approximate posterior (e.g., the eigendirections of the Fisher information or Hessian) offer a way to extend the intuition of Lemmas 3.3 and 3.5 to this setting? Or does the lack of tractable posterior geometry in the non-conjugate case make such a decomposition fundamentally unavailable?

---

> > > ### Author Response · Authors · 2026-04-04
> > >
> > > We would like to thank the reviewer for their continued engagement and apologise for misunderstanding Q3 in our initial rebuttal.
> > >
> > > The theory of our paper is specific to conjugate linear regression setting and we cannot provide any guarantees in non-conjugate settings. However, we do believe that the results presented in this paper are a special case of a more general rule. Unfortunately, as we discuss in more detail below, the maths  to understand the non-conjugate case will be significantly more complex and would go beyond the scope of this work.
> > >
> > >
> > > The non-conjugate setting becomes challenging for a couple of reasons. Most straightforwardly, the first step of our approach is to use the parametric expressions for the exact posterior and KL divergence between normal distributions to write a closed form distribution for the MFVI posterior. As there generally aren't parametric expressions for the exact posterior or the KL divergences to a Gaussian variational posterior, the naive approach to generalising our results would not work.
> > >
> > >
> > > The challenge with applying the geometric features of the loss landscape$^1$, such as the Fisher information or Hessian that the reviewer mentioned, is that the curvature is defined for an individual parameter value $\theta$, and the curvature can change from parameter value to parameter value. In contrast, the variational posterior, $q(\theta)$, is a distribution over multiple parameter values. This means understanding the optimal variational posterior requires considering an average of the geometric features of the posterior over the distribution $q(\theta)$. As is discussed in [1], this change in the curvature  causes the mean  and variance of the variational parameters to become coupled. More concretely, it means that the MFVI posterior and the exact posterior will not generally share the same mean prediction. Therefore,  the statements which we make about the changes in the variance of the prediction in the conjugate case would not be sufficient to characterise the difference in the predictions in the non-conjugate case. Instead, we would need to make statements about how both the mean, the variance, and possibly higher order moments of the variational posterior and exact posterior compare to one another jointly.
> > >
> > >
> > > However, while we cannot provide theoretical guarantees yet, we do believe that our results around over- and underestimtation will generalise in some fashion, and they will be specifically relevant for Bayesian Deep Learning. Both overestimation of uncertainty in distribution [2] and underestimation of uncertainty out of distribution [3] have previously been observed in MFVI-BNNs in narrow settings. Given that our theory predicts both of these effects in the linear case and the qualitative similarity between the loss landscapes, which we discussed previously and have given more details about in our response to ELpz, we believe that there is a good chance our theory will be relevant to the BNN setting.
> > >
> > >
> > > We are very happy to extend our discussion of the challenges of extending our work to non-conjugate settings, as we provided here in our new section discussing BNNs. Similarly, we are happy to call out that both predictive variance overestimation at the data locations [2] and underestimation away from the data locations [3] have been previously observed in MFVI-BNNs, and that we believe that our results for over- and underestimation transition with covariate shifts will generalise to that setting.
> > >
> > >
> > > Our opinion is that the surprisingly strong and counterintuitive results we are able to prove in the conjugate linear setting will be of interest to the community and merit publication. We hope that after these rebuttals the reviewer will share our view and consider recommending our work for publication.
> > >
> > >
> > > [1] Khan, M. E., & Rue, H. (2023). The Bayesian learning rule. https://arxiv.org/abs/2107.04562
> > >
> > > [2] Coker, Beau, et al. Wide mean-field Bayesian neural networks ignore the data. https://proceedings.mlr.press/v151/coker22a.html
> > >
> > > [3] Foong, A. Y., Li, Y., Hernández-Lobato, J. M., & Turner, R. E. (2019). 'In-Between'Uncertainty in Bayesian Neural Networks. https://arxiv.org/abs/1906.11537
> > >
> > >
> > > ---------
> > >
> > > $^1$ We assume that the phrase "geometric structure of the approximate posterior" used by the reviewer is referring to the geometric structure of the log-posterior as a function of the model parameters $\theta$. We apologise if the reviewer was referring to the ELBO as a function of the variational parameters and will briefly answer here: the optimal variational parameters are a first order stationary condition so the curvature of the ELBO w.r.t. the variational parameters does not determine the optimal variational posterior which we study in this paper.

---

### Official Review · Reviewer_ELpz · 2026-03-05

**Soundness:** 4
**Presentation:** 4
**Significance:** 3
**Originality:** 3
**Overall Recommendation:** 5
**Confidence:** 3

**Summary:**

This paper studies the predictive variance of mean field variational inference (MFVI). While the current consensus is that this variance tends to be underestimated, the authors state this is an incomplete understanding and theoretically show under the Gaussian model that predictive variance is rather overestimated in directions where training data concentrates. This theoretical result is supported by illustrative experiments and its link to the cold posterior effect discussed.

**Compliance With Llm Reviewing Policy:**

Affirmed.

**Final Justification:**

The presentation and soundness are excellent.

In the rebuttal, the authors mainly clarified the scope of their work, which addressed my main questions and concerns. Although the theoretical framework is limited to linear models, I still believe this is a strong paper that will benefit subsequent work and shed light on the under- and over-estimation issues in Bayesian learning.

Therefore, I recommend acceptance.

**Key Questions For Authors:**

Q1. Is the phenomenon of overestimation generalizable to other VI models where the underestimation of variance is observed?

Q2. Based on the theoretical results derived in this paper, could the authors think of a method/strategy to remedy the over-/underestimation problem?

**Limitations:**

No, please refer to weaknesses.

**Strengths And Weaknesses:**

(Strengths)

S1. This paper is primarily theoretical, and its theoretical framework consists of two parts. First, the authors show that predictive variance is underestimated in average, which is the traditional consensus, then they prove that predictive variance is rather overestimated in directions where training data concentrates. In addition to being sound, the proof and accompanying narrative convincingly convey the main message.

S2 The experiments throughout the paper adequately support its discussion and arguments and are illustrative.

S3. The result that overestimation of variance is occurring in important directions is a probably impactful result, although theory is limited to Gaussian MFVI. It deepens our understanding about this topic and may lead to novel and more effective strategies in subsequent work.

(Weaknesses)

W1. The authors mainly analyze the predictive variance of Gaussian MFVI. The generalization of the result to other VI settings is unclear.

W2. Moreover, this work is basically limited to the analysis of the problem of under/overestimation of variance and does not provide concrete potential methodologies to avoid and solve it.

W3 (minor) In equation (7), $w_{pq} $ is not defined in the main paper. Its definition should be added. The result of equation (12) may be emphasized under the form of lemma or theorem. l.270 (right), 1_P should be defined to avoid misunderstanding. In Appendices A.6 and A.7, some equations are not well-aligned (i.e., equations (57), (58), (64)).

---

> ### Author Rebuttal · Authors · 2026-03-30
>
> We thank the reviewer for their review, and provide responses to questions and concerns below.
>
> > W1. Generalization beyond Gaussian MFVI.
>
> We emphatically agree that our work needs to be generalised to other settings. However, we chose to study Gaussian MFVI because it is an incredibly common method. As such we believe that the results for this case are still important.
>
> > W3 (minor)
>
> Thank you for these. We will implement these changes for the next version of the paper.
>
> > Q1. Is the phenomenon of overestimation generalizable to other VI models where the underestimation of variance is observed?
>
> Our theory for overestimating variance for in-distribution predictions and underestimating variance for out-of-distribution  predictions relies on the linear setting, so we cannot provide guarantees for non-conjugate models. However, we do expect that our results will generalise to what we consider the most notable setting where MFVI overestimates uncertainty for in-distribution predictions: Bayesian Deep Learning.
>
> In response to similar questions being asked by several reviewers we have decided to include a new section in our paper, addressing whether this phenomenon could explain variance overestimation in Bayesian Deep Learning. We provide the Figure for this experiment [here](https://anonymous.4open.science/r/anonymous_fig-171F/fig7.png), and the text below.
>
> ### Section 6: MFVI underconfidence in BNNs
>
>
>
> While our theoretical analysis has focused on BLR, the CPE is primarily observed in BDL, where MFVI is also known to exhibit underconfident predictive distributions in certain settings [1]. A natural question is therefore whether the mechanism we identify here, the overestimation of predictive variance on in-distribution data, also operates in BNNs.
>
> A key prediction of our theory is that the MFVI approximation will perform worse when regression takes place in a high-dimensional parameter space but the data concentrates near a low-dimensional subspace. This follows from the conservation law (Lemma 3.5): the average ratio of MFVI to exact predictive variance across posterior eigenvectors is fixed at one, so as the number of uninformed directions grows, the overestimation along the few data-informed directions must become increasingly severe to compensate. It is revealing to consider the loss landscape implied by this structure. The Hessian of the log-posterior will be nearly flat in most directions, controlled only by the prior, and sharp in only the few directions where the data provides information. This bulk-plus-outliers Hessian structure of the loss which has been observed in Deep Learning [2], and this structure may be similarly pathological for MFVI in both cases.
>
> To provide preliminary evidence that this mechanism operates beyond conjugate models, we train two BNNs on 1D sinusoidal data using IVON[3], an MFVI optimiser for Deep Learning. [Figure 7](https://anonymous.4open.science/r/anonymous_fig-171F/fig7.png) compares a small network (depth 3, width 16) with a larger network (depth 2, width 512). Both networks were fit with full batch training, used a small learning rate ($10^{-6}$) and were trained for a long time to reach convergence ($10^6$ steps).
> Despite the mean of both networks fitting the data reasonably well, the larger network exhibits substantially wider credible intervals in the region where training data is observed. This mirrors the basis function regression result in Figure 5: increasing the dimensionality of the parameter space while keeping the data fixed leads to a higher predictive variance on in-distribution inputs.
>
> [1] Coker, Beau, et al. "Wide mean-field Bayesian neural networks ignore the data." https://proceedings.mlr.press/v151/coker22a.html
> [2]  Sagun, Levent, Leon Bottou, and Yann LeCun. "Eigenvalues of the hessian in deep learning: Singularity and beyond." https://arxiv.org/abs/1611.07476
> [3] Shen, Yuesong, et al. "Variational learning is effective for large deep networks." https://arxiv.org/abs/2402.17641
>
> > Q2 & W2. Remedies for prediction over-/underestimation.
>
> We agree that development of more concrete methodologies to address the predictive variance estimation from MFVI is important, and this will be the topic of future work.
>
> We would like to emphasise that we discuss a partial solution in Section 4, where we discuss cold posteriors which simply scale the MFVI posterior covariance by a scalar. If the goal is to make predictions on in-distribution data, practitioners should consider reducing the variance of the MFVI posterior by using a temperature $T<1$. Similarly, for out-of-distribution data predictions, using a warmer temperature is typically desirable. We subsequently validate this approach in Section 5 across both synthetic and real data.

---

> > ### Author Rebuttal · Reviewer_ELpz · 2026-04-01
> >
> > Thank you very much for the careful response. After reading the rebuttal, I find that my concerns have been well addressed. The authors provided satisfactory clarifications, and I will therefore maintain my score.

---

### Official Review · Reviewer_SwqZ · 2026-03-10

**Soundness:** 4
**Presentation:** 4
**Significance:** 4
**Originality:** 3
**Overall Recommendation:** 5
**Confidence:** 4

**Summary:**

This paper considers the setting of mean-field variational inference applied to a conjugate Bayesian linear regression setting. The authors find that mean-field VI posteriors overestimate uncertainty in certain directions while underestimating uncertainty along other directions. Crucially, the direction along which uncertainty is overestimated is along the direction of the first principal component of the data distribution, suggesting that the predictive variance for in-distribution samples is too large, whereas for out-of-distribution samples it may be too small. These findings suggest benefits of tempered / cold posteriors in various regimes, which are confirmed by experimental findings.

**Compliance With Llm Reviewing Policy:**

Affirmed.

**Final Justification:**

The authors have addressed my concerns in the rebuttal and the paper is in general quite strong. I think that illuminates an important aspect of mean-field variational inference that is not well known. I therefore recommend acceptance.

**Key Questions For Authors:**

A few additional comments:
1.. On page 1, I would personally not capitalize "Machine Learning", "Variational Inference", and so on.
2. Page 7, figure 5 caption contains a typo: "in-distribution. data, and lower temperatures..."
3. Page 7: replace "sinc(x)" with "\text{sinc}(x)" or something similar to avoid having italicized characters in math mode
4. There were an abundance of capitalization and other errors in the references. For example, "gaussian" --> "Gaussian". Make sure to use proper capitalization for journals, conferences, etc.

**Limitations:**

The authors can expand more on potential future work in the discussion section of the paper.

**Strengths And Weaknesses:**

The paper is technically sound and all theoretical results seem to be correct. Moreover, the technical derivations/intuition make the results easy to follow as one is reading. I also found that the paper was very clearly written and well structured, with figures throughout the manuscript helping to further reinforce understanding.

In my opinion, this paper addresses an important problem, as it is well understood that MFVI typically underestimates uncertainty, however it is not well acknowledged that it overestimates uncertainty along crucial directions corresponding to the primary principal component of the training data. I found this work both insightful and practical.

Here are a few more additional comments:
- I appreciated Figure 1 and found that it was a useful reference point when reading the rest of the paper.
- I think that Theorem 3.6 can be alluded to earlier as it is quite an interesting result.
- Can you add some discussion to Figure 6 why $T < 1$ is preferred on the joint divergences for W2 and Frobenius norm on OOD data, unless if I have missed that discussion elsewhere.

---

> ### Author Rebuttal · Authors · 2026-03-29
>
> We would like to thank the reviewer for the nice review.
>
> > I think that Theorem 3.6 can be alluded to earlier as it is quite an interesting result.
>
> Thank you for this comment. After l65 LHS, discussing the toy example we will add the following line:
>
> "Our main result, given in Theorem 3.6, is to show that the overestimation of predictive variance at the training data in MFVI linear regression is very general. Specifically, if a spherical prior is used, then MFVI will always overestimate the exact posterior predictive variance for the training data."
>
> > Can you add some discussion to Figure 6 why $T<1$ is preferred on the joint divergences for W2 and Frobenius norm on OOD data, unless if I have missed that discussion elsewhere.
>
> Thank you for pointing out this oversight in the discussion of our results. OOD predictions can still benefit from a temperature of $T<1$ if the covariate shift is small. For example, in the Toy Example in Figure 1, there is a small region around the data subspace in which the variance of the MFVI prediction still overestimates the exact predictive uncertainty. If the OOD dataset was drawn from that region, a value of $T<1$ may still match the exact posterior prediction more closely than using $T\geq 1$. In our UCI dataset example, the precise optimal temperature will depend on the learned hyperparameters of the basis functions, the precise distributions of the ID and OOD datasets in the feature space, and the divergence used to compare the predictions.
> The important relationship is that the OOD predictions require a warmer temperature than the ID predictions, which is true in all cases we examine. We will clarify our discussion on this point.
>
> > A few additional comments: 1.. On page 1, I would personally not capitalize "Machine Learning", "Variational Inference", and so on. 2. Page 7, figure 5 caption contains a typo: "in-distribution. data, and lower temperatures..." 3. Page 7: replace "sinc(x)" with "\text{sinc}(x)" or something similar to avoid having italicized characters in math mode 4. There were an abundance of capitalization and other errors in the references. For example, "gaussian" --> "Gaussian". Make sure to use proper capitalization for journals, conferences, etc.
>
> Thank you for your care in identifying these errors, we will ensure they are corrected for future versions.

---

> > ### Author Rebuttal · Reviewer_SwqZ · 2026-04-02
> >
> > Thank you for your responses to my questions. I believe all of my concerns have been fully resolved and I am maintaining my score.

---

### Official Review · Reviewer_5koy · 2026-03-13

**Soundness:** 3
**Presentation:** 3
**Significance:** 2
**Originality:** 3
**Overall Recommendation:** 4
**Confidence:** 3

**Summary:**

This paper studies Mean Field Variational Inference (MFVI) from the perspective of predictive uncertainty. Focusing on conjugate Bayesian linear regression, the authors show that although MFVI underestimates uncertainty in parameter space, it can overestimate predictive variance along certain directions in the input space, and they characterize this behavior through the structure of the MFVI covariance and its relation to the geometry of the training data. The numerical study illustrates the theory through synthetic linear regression, basis function regression, and experiments on several UCI regression datasets.

**Compliance With Llm Reviewing Policy:**

Affirmed.

**Final Justification:**

After considering both the paper and the rebuttal, I find that my concerns have been properly addressed, and the rebuttal has strengthened my confidence in the paper’s soundness, significance, and overall contribution.

**Key Questions For Authors:**

- The theoretical analysis in the paper focuses on conjugate Bayesian linear regression with Gaussian priors and likelihoods. Could the authors comment on how the identified phenomenon might extend to more realistic nonlinear models, such as Bayesian neural networks where MFVI is commonly used? In particular, do the authors expect the predictive variance inflation along data-aligned directions to persist in deep neural networks?

- Several prior works have argued that performing variational inference in parameter space can lead to distortions in predictive uncertainty and have proposed performing inference directly in function space instead, for example through functional variational inference methods. Could the authors clarify how the phenomenon described in this paper relates to these observations? In particular, is the predictive variance inflation identified here one of the mechanisms motivating function-space approaches?

- The experiments mainly illustrate the theoretical results using synthetic regression problems, basis function regression, and UCI datasets. Do the authors have evidence that the same predictive variance behavior occurs in more complex models, such as deep neural networks trained with MFVI? Such experiments could help clarify whether the proposed explanation for the Cold Posterior Effect is relevant in practical deep learning settings.

- The paper proposes predictive variance overestimation as a possible explanation for the Cold Posterior Effect. However, the literature also contains several other explanations, such as likelihood misspecification, data augmentation, or prior misspecification. Could the authors discuss how their explanation interacts with these previously proposed mechanisms, and whether they expect the predictive variance effect identified here to be a dominant factor in practice?

**Limitations:**

Partially. The paper briefly acknowledges a practical limitation, namely that the mismatch between parameter-space and predictive uncertainty becomes especially pronounced in high-dimensional or low-rank settings, but this could be discussed more explicitly as a limitation of the scope and applicability of the analysis.

**Strengths And Weaknesses:**

Strengths

- **Conceptually interesting perspective on MFVI and predictive uncertainty.**
  The paper provides a clear analysis of the predictive behavior of Mean Field Variational Inference (MFVI). While it is well known that MFVI tends to underestimate posterior variance in parameter space, the paper highlights that this intuition does not necessarily carry over to predictive uncertainty. In particular, the results show that MFVI can overestimate predictive variance along directions aligned with the training data, including for test points drawn from the empirical distribution of the training inputs.

- **Clean and interpretable theoretical analysis.**
  By focusing on conjugate Bayesian linear regression, the authors derive several interpretable results characterizing the relationship between the exact posterior covariance and the MFVI covariance. In particular, the harmonic-mean structure of the MFVI eigenvalues provides useful intuition for why predictive variance may be inflated along certain directions. The mathematical arguments are clear and relatively easy to follow.

- **Interesting connection to the Cold Posterior Effect.**
  The paper relates its predictive variance analysis to the Cold Posterior Effect (CPE), suggesting that tempering the posterior can correct predictive variance distortions induced by MFVI. Although the setting is simplified, this provides a useful perspective on why colder posteriors may sometimes yield improved predictive performance.

Weaknesses

- **The scope of the theoretical analysis is limited.**
  The results are derived in the setting of conjugate Bayesian linear regression with Gaussian priors and likelihoods. While this allows for a clean theoretical treatment, it remains unclear to what extent the phenomena described in the paper generalize to more realistic models such as Bayesian neural networks or other nonlinear settings where MFVI is commonly applied. Since the Cold Posterior Effect is primarily discussed in deep learning models, the connection between the presented theory and practical deep learning settings remains somewhat indirect.

- **The empirical evaluation is relatively limited.**
  The experiments focus mainly on synthetic examples, basis function regression, and several UCI regression datasets. These experiments illustrate the theoretical findings, but they do not include modern deep neural networks or large-scale benchmarks. As a result, it remains unclear whether the predictive variance behavior identified in the analysis plays a significant role in the practical settings where MFVI and cold posteriors are typically studied.

- **The novelty of the contribution appears moderate.**
  Prior work has already analyzed the behavior of MFVI approximations in parameter space and noted that variational approximations can distort uncertainty estimates. The main contribution here is to characterize how such distortions manifest in predictive variance and to relate them to the Cold Posterior Effect. While this perspective is interesting and clearly presented, the overall conceptual advance may be viewed as somewhat incremental relative to the broader literature on MFVI uncertainty distortions.

---

> ### Author Rebuttal · Authors · 2026-03-30
>
> We thank the reviewer for their review, and provide responses to questions and concerns below.
>
> > W1. The scope of the theoretical analysis is limited.
>
> The results we present in this paper are calling out to be generalised to non-conjugate settings and this will be the focus of future work. However, the surprisingly strong results in the linear regression case, which we have presented in this paper, are still an important theoretical development.
>
> > W3. The novelty of the contribution appears moderate.
>
> We respectfully disagree. Prior work on MFVI distortions has consistently concluded that MFVI underestimates uncertainty. Our result is a counter-intuitive reversal: for the very common setting of in-distribution predictions with spherical priors, the uncertainty is overestimated. This is a reversal in the direction of the distortion, rather than an incremental update of the existing understanding.
>
> If the reviewer was thinking of specific references which would make ours incremental, we would welcome them and are happy to deal with them directly. Otherwise, we will extend our discussion with the above points to emphasise the reasons we feel our work is novel and significant.
>
> > Q1, Q3. Extending results to Deep Learning &  W2.  The empirical evaluation is relatively limited.
>
> The experiments we included were primarily to validate and explain our findings for linear regression, and we hope that the reviewer agrees that they achieve this goal. However, in response to this and other comments, we have decided to include discussion and simple Deep Learning example in our paper. We have provided the text and links to the Figure in our response to Reviewer ELpz, but a brief justification is given here too.
>
> The examples in which we demonstrate MFVI severely overestimates the predictive variance share a similar geometry to the loss-landscape of deep neural networks. Specifically, the examples where we show that linear regression significantly overestimates predictive variance have a high curvature of the log-posterior density in only a few directions and low curvature in a much greater number. This Hessian structure is exactly the one which is present in Deep Learning [5].
>
> We will include a demonstration of a similar behaviour to the basis function regression in a new [Figure 7](https://anonymous.4open.science/r/anonymous_fig-171F/fig7.png). In it we show that, similar to Figure 5, as the number of parameters of the model is increased, the predictions at the data locations become less confident for MFVI.
>
> We hope that including this additional discussion and result will convince the reviewer that our research is relevant to deep learning.
>
> [5] Sagun, Levent, Leon Bottou, and Yann LeCun. "Eigenvalues of the hessian in deep learning: Singularity and beyond." https://arxiv.org/abs/1611.07476
>
> > Relation of functional VI.
>
> In the linear regression setting we study, there is a one-to-one mapping between parameters and functions, so the function-space and parameter-space VI objectives are identical (see [2], Proposition 3) and functional VI would not avoid the pathology we describe.
>
> In deep learning, functional VI is typically motivated by placing interpretable GP priors over predictions [1,2], rather than by correcting predictive variance distortions of the kind we identify. In the GP literature, function-space methods such as inducing-point approaches [3,4] are primarily introduced to improve computational complexity.
>
> We will expand our discussion of functional VI in Section 2 accordingly.
>
> [1] Sun, Shengyang, Guodong Zhang, Jiaxin Shi, and Roger Grosse. ‘Functional Variational Bayesian Neural Networks’.  https://doi.org/10.48550/arXiv.1903.05779.
>
> [2] Burt, David R., Sebastian W. Ober, Adrià Garriga-Alonso, and Mark van der Wilk. ‘Understanding Variational Inference in Function-Space’. https://doi.org/10.48550/arXiv.2011.09421.
>
> [3] Titsias, Michalis. ‘Variational Learning of Inducing Variables in Sparse Gaussian Processes’. https://proceedings.mlr.press/v5/titsias09a.html.
>
> [4] Hensman, James, Nicolo Fusi, and Neil D. Lawrence. ‘Gaussian Processes for Big Data’. https://doi.org/10.48550/arXiv.1309.6835.
>
> > Q4. Relation to existing CPE effects
>
> As the reviewer notes, there has been excellent prior work attributing the CPE to different explanations. Our work shows that, in addition to the existing explanations, a CPE can arise purely from correcting distortions introduced by MFVI, making predictions closer to those from the exact Bayesian posterior. We believe this mechanism is novel in the CPE literature.
>
> None of the explanations, including ours, are mutually exclusive and can all be present at once. Which factor is dominant will depend on the precise model and data. In cases where the likelihood is well specified and there is no data augmentation, we do expect that the effect described in this work will be significant.

---

> > ### Author Rebuttal · Reviewer_5koy · 2026-04-04
> >
> > Thank you for your detailed response. My questions have been adequately addressed. I will maintain my original score, which is already on the positive (accept) side.

---

### Decision · Program_Chairs · 2026-04-30

**Decision:**

Accept (regular)

**Comment:**

This paper analyzes Gaussian mean-field variational inference (MFVI) in Bayesian linear regression and shows that, contrary to the common belief that MFVI uniformly underestimates uncertainty, it can overestimate predictive variance in directions aligned with the training data. This effect is characterized through a clean decomposition of the MFVI covariance structure and its interaction with data geometry, and is further connected to the Cold Posterior Effect. The theoretical results are supported by illustrative synthetic and UCI regression experiments.

Across all four reviews, there is strong agreement that the paper is technically sound, clearly written, and based on a clean and insightful theoretical analysis. Reviewers consistently highlight the clarity of the derivations and the usefulness of the geometric interpretation of MFVI predictive behavior. The main contribution is viewed as a meaningful refinement of the standard understanding of MFVI uncertainty, particularly by distinguishing parameter-space underestimation from direction-dependent predictive overestimation.

The main concerns are also consistent: the analysis is limited to Gaussian MFVI in linear regression, with unclear generalization to nonlinear models such as Bayesian neural networks. The empirical evaluation is relatively narrow and does not include modern deep learning benchmarks. In addition, the paper is primarily diagnostic, offering limited methodological innovation beyond temperature scaling, and does not fully resolve how to choose or optimize such corrections in practice.

The rebuttal strengthens the submission by clarifying the scope of the work as theoretical, providing additional discussion and preliminary neural network experiments suggesting similar behavior in wider models, and further motivating the connection to Hessian structure in deep learning. It also clarifies that temperature scaling is intended as a practical heuristic rather than a fully principled solution, and emphasizes that the observed effect differs between in-distribution and out-of-distribution settings.

Overall, the reviewers converge on a positive assessment, and the paper is recommended for acceptance as a clear and insightful theoretical contribution that refines the understanding of MFVI uncertainty, despite its limited scope and empirical coverage.